# Seeing is Understanding: Unlocking Causal Attention into Modality-Mutual Attention for Multimodal LLMs

## Abstract

Recent Multimodal Large Language Models (MLLMs) have demonstrated significant progress in perceiving and reasoning over multimodal inquiries, ushering in a new research era for foundation models. However, vision-language misalignment in MLLMs has emerged as a critical challenge, where the textual responses generated by these models are not factually aligned with the given text-image inputs. Existing efforts to address vision-language misalignment have focused on developing specialized vision-language connectors or leveraging visual instruction tuning from diverse domains. In this paper, we tackle this issue from a fundamental yet unexplored perspective by revisiting the core architecture of MLLMs. Most MLLMs are typically built on decoder-only LLMs consisting of a causal attention mechanism, which *limits the ability of the earlier modalities (e.g., images) to incorporate information from the latter modalities (e.g., text)*. To address this problem a MLLM that unlocks causal attention into our proposed modality-mutual attention (MMA) to enable image tokens to attend to text tokens. This simple yet effective design allows MMA to achieve state-of-the-art performance in 12 multimodal understanding benchmarks (**+6.2%** on average across **3** LLMs backbones) without introducing additional parameters. Our MMA design is intended to be generic, allowing for applications across various modalities, and scalable to accommodate diverse multimodal scenarios[1].

## 1 Introduction

Recently, Multimodal Large Language Models (MLLMs) have demonstrated remarkable capabilities in multimodal understanding (Anil et al., 2023; McKinzie et al., 2024; Liu et al., 2024b; Xue et al., 2024; Li et al., 2024), paving the way for innovative applications in general-purpose foundation models (Yin et al., 2023). By leveraging visual instruction tuning (Liu et al., 2023b) with large-scale vision-text datasets spanning diverse domains (e.g., Visual Question Answering (VQA), Optical Character Recognition (OCR), and coding), MLLMs can generate coherent and contextually accurate responses to user queries involving both visual and textual inputs (Abdin et al., 2024; Wu et al., 2024). Despite these advancements, a significant concern limiting the reliability and applicability of MLLMs is the issue of object hallucinations, where the generated outputs are not factually aligned with the provided inputs (Liu et al., 2024a). This limitation becomes especially evident in tasks demanding robust cross-modal interactions, such as vision-centric tasks (e.g., object counting, spatial relations) requiring accurate object-location descriptions. Figure 1 illustrates a vision-centric paradigm that has ambiguous signs in the image. Both the proprietary model (GPT-4o (OpenAI, 2024)) and open-source models (Molmo (Deitke et al., 2024), and DeepSeek-VL2-Small (Wu et al., 2024)) suffer from object hallucinations by responding to the user query with inaccurate *No Parking* and *Passenger Loading* signs.

Previous research has mitigated the vision-language misalignment issue through various approaches. One promising direction involves data-centric approaches, which scale up the diversity of vision-related instruction data during the supervised finetuning stage. For instance, Molmo (Deitke et al., 2024) incorporates not only conventional academic datasets but also specialized datasets, such as

---

[1]Code and model will be available in the camera-ready version.

Figure 1: An illustration of the vision-centric scenario. The image contains ambiguous signs with the object-related query. The correct answer is that parking is allowed for 2 hours from 8am to 8pm on Saturday. While GPT-4o (OpenAI, 2024), Molmo (Deitke et al., 2024), and DeepSeek-VL2-Small (Wu et al., 2024) respond with hallucinations, our proposed AKI is able to provide an accurate answer. The image is sourced from (Sanders, 2015).

clocks, pointing, and counting tasks, to ground visual understanding. LLaVA-1.5 (Liu et al., 2024b) and MM-1.5 (Zhang et al., 2024) also demonstrate the effectiveness of multimodal understanding by expanding the range of visual instruction samples. Another key direction focuses on vision-language connectors, which aim to improve the alignment between visual and textual representations. Cha et al. (2024) introduced abstractors that provide both flexibility and locality preservation, while Tong et al. (2024) proposed spatial vision aggregators that explicitly define the aggregation space for each query token and repeatedly aggregate vision features across LLM layers. Nonetheless, the challenge of scaling training data remains a significant barrier for researchers with limited resources. Furthermore, design choices for vision-language connectors have yet to reach a consensus on their optimal implementation (McKinzie et al., 2024).

In this paper, we tackle the misalignment issue from a fundamental yet unexplored perspective by revisiting the major component of MLLMs. As illustrated in Figure 2, most existing MLLMs are built on top of LLMs that incorporate the causal attention mechanism, which was originally designed to prevent earlier text tokens from attending to future text tokens in autoregressive text generation. Taking a single image-text pair as an example, to accommodate multimodal input for the LLM, the input sequence is typically formed by placing image tokens before text tokens. However, this design introduces a critical limitation: the former modality (images) cannot attend to or incorporate information from the latter modality (text), thereby hindering effective cross-modal interactions. This limitation raises a fundamental question: *How can we enable the former modality to effectively consider and integrate information from the latter modality in MLLMs?*

An intuitive solution to this problem involves directly pretraining and finetuning an MLLM using both image-text and text-image input orders, which can be easily implemented within existing training pipelines and enhances robustness by exposing the model to different input sequences. While this method shows improvements on some multimodal understanding benchmarks (see our experiments in Section 4.3 for details), it doubles the required training time for each stage and incurs exponential growth in training costs as the number of modalities increases. To address this, we propose a novel MLLM that redefines the causal attention mechanism into modality-mutual attention (MMA) within the LLM. Specifically, attention masks in the LLM are modified to allow image tokens to attend to the question text tokens during the supervised finetuning stage. This *unlock* strategy offers multiple key advantages: 1) image tokens gain the ability to access and interpret user queries; 2) it solely modifies the attention masks within the LLM, enabling seamless integration into existing MLLM training frameworks; and 3) no additional parameters are introduced.

Compared to the conventional causal-attention framework, state-of-the-art baselines, and our intuitive dual-order method, all using the same training data and configurations, our simple yet effective

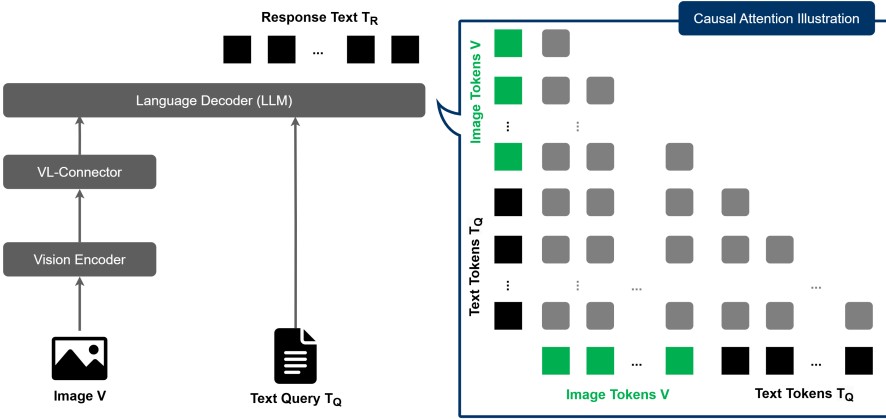

Figure 2: The conventional framework for MLLMs (e.g., Molmo (Deitke et al., 2024), BLIP-3 (Xue et al., 2024), and Cambrian (Tong et al., 2024)) typically consists of a vision encoder, a vision-language (VL) connector, and a text decoder (LLM). In this framework, images are often placed before text in a sequentialized input, causing the former modality (images) lacking access to information from the later modality (text) due to the causal attention design in decoder-only LLMs, as shown in the right part (gray squares). Notably, placing text before image tokens does not resolve this issue, as the fundamental limitation persists.

MMA approach consistently demonstrates significant improvements across 12 multimodal benchmarks (**+6.2%** average performance gain compared to the state-of-the-art baselines across **3** LLM backbones), including vision-centric, knowledge-based, and general tasks.

Our contributions are summarized as three-fold: **Problem Novelty:** We address the vision-language misalignment issue by revisiting the foundational framework of MLLMs and identifying a critical limitation in the causal attention mechanism of LLMs when processing multimodal inputs. Specifically, we highlight that the sequential input design prevents the earlier modality from accessing information from the latter modality. **Method Novelty:** We first explore an intuitive approach that demonstrates the importance of enabling cross-modal information flow, though it incurs additional training costs. Therefore, we propose modality-mutual attention (MMA) to mitigate object hallucinations by unlocking the attention masks to allow tokens from the former modality to attend to contextual information from the latter modality. **Moreover, we conduct a systematic analysis of MMA to provide guidance on how image tokens attend to text tokens across different scenarios.** **Experiment Novelty:** Through extensive evaluations on 12 multimodal understanding benchmarks across **3** LLMs, our MMA approach consistently outperforms state-of-the-art models and carefully designed variant baselines, demonstrating its effectiveness in advancing multimodal reasoning and understanding.

## 2 RELATED WORKS

**Multimodal Large Language Models.** The rapid development of MLLMs has been witnessed with their capabilities to comprehend and process multimodal information, reinforcing them to perceive, reason, and respond across diverse applications (Yin et al., 2023). Building on earlier milestones, such as Flamingo (Alayrac et al., 2022), which pioneered the integration of visual and textual information, recent advancements (e.g., LLaVA-1.5 (Liu et al., 2024b), QWen-VL (Bai et al., 2023), GPT-4o (OpenAI, 2024), Phi-3-Vision (Abdin et al., 2024), and Claude 3.5-Sonnet (Anthropic, 2024)) have employed visual instruction tuning (Liu et al., 2023b) to imbue models with domain-specific expertise, enhancing their ability to ground visual information effectively. Generally, as illustrated in Figure 2, most MLLMs share a standard architecture comprising three core components (Caffagni et al., 2024): a vision encoder (often a CLIP-style) to extract image features, a vision-language (VL) connector to align visual embeddings with textual representations, and a text decoder (LLM) to reason over the combined image-text inputs and serve as the user-facing inference interface. Despite these substantial advancements, the misalignment issue between visual contents

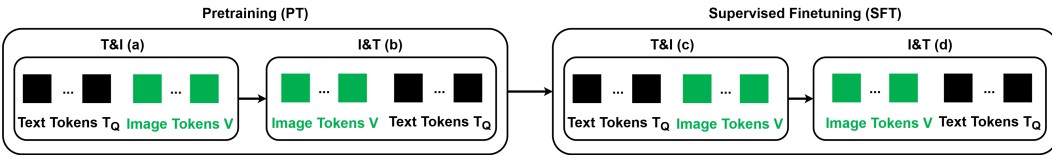

Figure 3: An illustration for dual-order training, where T and I indicate text and images.

and textual responses remains an emerging challenge, which often results in inaccurate or inconsistent outputs by heavily depending on the knowledge of LLMs, posing a significant hurdle, especially to the vision-centric applications (Tong et al., 2024).

**Vision-Language Misalignment.** Existing works addressing the misalignment issue in end-to-end manners can be mainly categorized into two strategies. The first category comprises data-centric approaches, which focus on collecting diverse visual instruction tuning samples by annotating images with vision models or human input, followed by generating question-answer pairs using LLMs (You et al., 2024; Zhu et al., 2024; Dong et al., 2025; Deitke et al., 2024). Although these methods enhance the fine-grained grounding capabilities of MLLMs, scaling data for training MLLMs remains a costly endeavor, especially for researchers with limited computational resources. The other alternative involves model-centric methods, which mainly aim to improve the vision-language connector, an important component for aligning vision and text representations. Common approaches include resamplers (Alayrac et al., 2022), Q-Formers (Li et al., 2023b), and linear projectors (Liu et al., 2023b), all of which are widely adopted in many MLLMs. Recently, locality-preserved abstractors (Cha et al., 2024) have demonstrated superior performance compared with earlier connectors; similarly, Cambrian (Tong et al., 2024) introduced a spatial vision aggregator by explicitly defining the aggregation space and incorporating vision features across LLM layers. However, as empirically experimented in (McKinzie et al., 2024), no single vision-language connector consistently outperforms across diverse multimodal benchmarks, leaving their optimal design an open question.

From a more foundational perspective, we explore the misalignment issue by revisiting the core architecture of LLMs, where causal attention prevents the earlier modality (e.g., images) from attending to the later modality (e.g., text). Mixed attention (Xie et al., 2025) is introduced in the unified multimodal Transformers for understanding and generation, which enables full attention across image tokens yet remaining no information from image to text tokens. The concept of concentric causal attention (CCA) (Xing et al., 2024) is closely related to our work, as it manipulates position encoding and attention masks for image tokens to prevent hallucinations; nonetheless, CCA assumes central regions of an image are more critical, and still suffers from the cross-modal interaction issue. Our proposed modality-mutual attention, on the flip side, unlocks causal attention between modalities for cross-modal interactions, achieving effective alignment performance across extensive multimodal benchmarks.

## 3 HOW TO ENABLE THE FORMER MODALITY TO SEE THE LATTER MODALITY?

### 3.1 OVERVIEW

Generally, we follow the standard pipeline illustrated in Figure 2, which first divides an image into multiple patches that are fed into the vision encoder $f_V$ to produce vision representations. Then, the LLM $f_L$ integrates these representations, which are first processed by the vision-language connector $p_V$, with text representations encoded from the text query – e.g., captions in the pretraining (PT) stage, question-answer pairs in the supervised finetuning (SFT) stage – to generate a response autoregressively. Formally, given an image-text pair $S = [V, T_Q]^2$, where the image $V$ appears first, followed by the text query $T_Q$, with the corresponding length $|V|$ and $|T_Q|$, respectively, the objective is to generate a response $T_R$:

$$T_R = f_L(H_V, H_{T_Q}), \text{where } H_V = f_V(p_V(V)) \in \mathbb{R}^{|V| \times d}; H_{T_Q} = f_T(T_Q) \in \mathbb{R}^{|T_Q| \times d}, \quad (1)$$

---

[2]In the SFT stage, a system message is placed before $V$ similar to (Cha et al., 2024). We omit the system text tokens in this section for better readability.

where $f_T$ is the text token embedder in the LLM and $d$ is the hidden dimension of the LLM.

To enforce autoregressive generation, where tokens are only allowed to have access to their corresponding previous tokens, LLMs are built using decoder-only Transformers with causal attention and feed-forward networks (Vaswani et al., 2017). Specifically, causal attention can be formulated:

$$Attention_{causal} = softmax(\frac{QK^T + M}{\sqrt{d}}), \text{where } \mathbf{M}_{ij} = \begin{cases} 0 & \text{if } j \leq i, \\ -\infty & \text{if } j > i, \end{cases} \tag{2}$$

where $M$ is the triangular indicator matrix ensuring that only positions $j \leq i$ are attended to by applying $-\infty$ to disallowed positions.

## 3.2 INTUITIVE APPROACH: DUAL-ORDER TRAINING (DOT)

The conventional pipeline for training an MLLM follows the I&T order in both the PT and SFT stages (i.e., only (b) and (d) in Figure 3), where image tokens precede text tokens in the sequence $S = [V, T_Q]$. To mitigate modality blindness, we introduce a dual-order training (DOT) method that further trains the model using the T&I order, $\hat{S} = [T_Q, V]$. As illustrated in Figure 3, we opt for a tandem training pipeline, where the model is first trained with the T&I order, followed by the I&T order in each stage, ensuring alignment with the I&T order during inference:

> **Prompt Template for the I&T and T&I Orders**
>
> // 1. I&T
> `<image>`
> {image patch$_1$}, $\cdots$, {image patch$_{|V|}$}
> {question}
> // 2. T&I
> {question}
> `<image>`
> {image patch$_1$}, $\cdots$, {image patch$_{|V|}$}

Figure 4: The prompt template for the I&T and T&I input orders. {image patch} and {question} are replaced based on each data sample.

$$\text{Conventional: } [V_{PT}, T_{Q_{PT}}] \rightarrow [V_{SFT}, T_{Q_{SFT}}]. \tag{3}$$

$$\text{DOT: } [T_{Q_{PT}}, V_{PT}] \rightarrow [V_{PT}, T_{Q_{PT}}] \rightarrow [T_{Q_{SFT}}, V_{SFT}] \rightarrow [V_{SFT}, T_{Q_{SFT}}]. \tag{4}$$

This strategy learns robust representations by incorporating not only the standard I&T but also T&I orders during training. In addition, it provides a plug-and-play advantage, as it merely requires reordering text and image inputs before feeding them into the model. Nonetheless, compared with the conventional pipeline (Equ. 3), a key drawback is the increased computational cost as shown in Equ. 4 due to the need for training both orders, requiring twice the training time per stage. Note that this overhead scales factorially to $n!$ when the input consists of $n$ modalities. The prompt template for incorporating I&T and T&I orders is described in Figure 4.

## 3.3 MODALITY-MUTUAL ATTENTION (MMA)

When incorporating multimodal inputs, most open-source and proprietary MLLMs (e.g., MM-1.5 (Zhang et al., 2024), LLaVa-1.5 (Liu et al., 2024b)) follow earlier approaches (e.g., Flamingo (Alayrac et al., 2022)) by positioning images before text in the input sequence. While this I&T order allows text to reference visual information, it blocks the path for images to see textual information due to causal attention in the LLM, which may be one of the reasons why MLLMs are prone to vision-centric tasks (Tong et al., 2024). For instance, when describing the object count or location as a user query, the model would need the text input (perhaps including object names or relations) to inform its understanding of the image. However, causal attention, which is designed to prevent information leakage in unimodal settings, leads

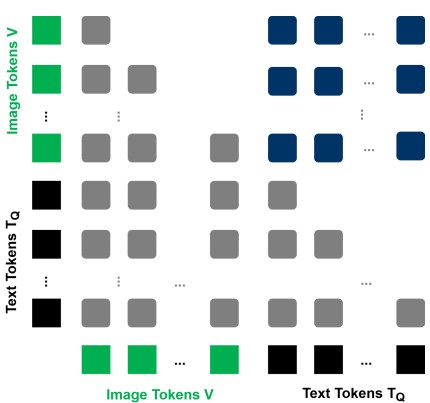

Figure 5: An illustration for our proposed modality-mutual attention (MMA), which modifies the causal attention mask in the LLM (gray squares) by enabling the information flow from image tokens to text tokens (blue squares).

to static representations regardless of question

changes. Moreover, simply reversing the modality order (i.e., T&I) does not resolve the issue, as text tokens still cannot attend to image tokens.

To this end, we propose modality-mutual attention that alters the causal attention mechanism in the LLM. As illustrated in Figure 5, our approach replaces the standard causal attention mask by unlocking the paths that allow image tokens to attend to text tokens, all without requiring the LLM retraining from scratch. Formally, $M$ in Equ. 2 can be devised as $M'$:

$$\mathbf{M}'_{ij} = \begin{cases} 0 & \text{if } j \leq i, \\ 0 & \text{if } 1 \leq i \leq |V| \text{ and } |V| + 1 \leq j \leq |V| + |T_Q|, \\ -\infty & otherwise. \end{cases} \tag{5}$$

In this manner, image tokens are able to learn to focus on regions linked to question tokens. Note that the MMA design merely unlocks some of the attention masks from $\mathbf{M}$ to $\mathbf{M}'$, which therefore does not increase computational cost as the total number of calculating 0 and $-\infty$ remains the same (i.e., $(|V| + |T_Q|)^2$).

Furthermore, our approach naturally extends to interleaved multimodal data, where greater cross-modal visibility is essential for capturing rich multimodal interactions. To accommodate such cases, we propose a generalized condition:

$$\mathbf{M}'_{ij} = \begin{cases} 0 & \text{if } j \leq i \text{ or } \phi(i) \neq \phi(j), \\ -\infty & \text{if } otherwise, \end{cases} \tag{6}$$

where $\phi(i)$ maps $i$-th token to its corresponding modality (e.g., audio, image, text).

The modality-mutual attention is applied with the given input $S$ to be stored in the KV cache – the generated tokens (i.e., $T_R$) employ the standard causal attention as in Equ. 2. In this paper, we focus on applying our modality-mutual attention during the SFT stage since the pretraining stage aims at understanding the context of the image-text pairs; there is no specific user instruction or questions that can specify image tokens to attend to (e.g., "How many cars are in the image?").

### 3.4 TRAINING

To ensure fair comparisons, we train both the DOT and MMA methods using a standard two-stage pipeline, consisting of pretraining and supervised finetuning. During both stages, we freeze the vision encoder and unfreeze the VL-connector and LLM to enhance the model's capability in vision-text alignment and multimodal understanding. We train the LLM with all parameters instead of parameter-efficient finetuning. To learn vision-text alignment, Blip3-kale (Awadalla et al., 2024) is adopted as the pretraining captioning dataset to understand how visual signals align with textual captions. Following pretraining, the supervised finetuning stage aims to equip the model with instruction-following abilities while further improving its capacity for diverse multimodal comprehension. Similar to (Cha et al., 2024), we incorporate a range of datasets to cover different multimodal reasoning tasks: 1) VQAv2 (Goyal et al., 2017), VSR (Liu et al., 2023a), GQA (Hudson & Manning, 2019), and OCRVQA (Mishra et al., 2019) for learning open-ended VQA; 2) ScienceQA (Lu et al., 2022) and A-OKVQA (Schwenk et al., 2022) for learning multi-choice VQA; 3) Ref-COCO (Kazemzadeh et al., 2014), RefCOCOg (Mao et al., 2016), RefCOCO+ (Yu et al., 2016), and VisualGnome (Krishna et al., 2017) for referring expression comprehension (visual grounding and grounded captioning); and 4) Llava-150k (Liu et al., 2023b) for visual instruction-following. Detailed statistics, dataset descriptions, and templates including instructions are depicted in Appendix C.2 and D.

## 4 EXPERIMENTS

### 4.1 EXPERIMENTAL SETUP

**Benchmarks.** We evaluate MMA on a diverse collection of 12 multimodal understanding benchmarks, which can be categorized into 3 groups to assess multifaceted capabilities.

**1) General:** MME (Fu et al., 2023) assesses perception and cognitive abilities across multiple subtasks. We denote its perception and cognition splits as $MME^P$ and $MME^C$, respectively. MMBench

(MMB) (Liu et al., 2024c) evaluates answer robustness by applying extensive shuffling to multiple-choice answers. SEED-Bench (Li et al., 2023a) is a multi-choice VQA task covering multifaceted understanding. We evaluate its image-based subset, denoted as $SEED^I$. LLaVA-Bench ($LLaVA^W$) (Liu et al., 2023b) assesses the correctness and helpfulness of visual conversations across diverse tasks using LLM-as-a-judge.

**2) Knowledge:** MMMU (Yue et al., 2024) measures college-level perception and reasoning with domain-specific knowledge. MathVista (Lu et al., 2024b) evaluates mathematical reasoning in visual contexts. We use the test-mini set following (Xue et al., 2024), denoted as $MathV^{mini}$.

**3) Vision-Centric:** POPE (Li et al., 2023c) measures object hallucination in multimodal models. MM-Vet (Yu et al., 2024) examines MLLMs with multidisciplinary, complex questions and employs LLMs to evaluate the quality of open-ended responses. RealworldQA (xAI, 2024) focuses on vision-centric tasks, evaluating spatial understanding from real-world images. CV-Bench (Tong et al., 2024) assesses vision-centric capabilities, including 2D spatial relationships and object counting ($CV\text{-}Bench^{2D}$) as well as 3D depth ordering and relative distance ($CV\text{-}Bench^{3D}$).

**Evaluation metrics.** The evaluation metrics for each benchmark are computed using official implementations by default, and all experiments are conducted in a 0-shot manner. Except for CV-Bench, we leverage the VLMEval toolkit (Duan et al., 2024) to fairly evaluate the baselines and our MMA approach. In terms of CV-Bench, we report 2D and 3D accuracy separately following the official evaluation codes. All the results are the average of 3 different random seeds.

**Implementation details.** All experiments adopt Phi-3.5-Mini-Instruct (Abdin et al., 2024) and LLaMA-3.2-3B-Instruct (Meta, 2024), respectively, as the pretrained LLMs to examine the generalizability of our MMA, the Perceiver Resampler (Alayrac et al., 2022) as the VL connector to map visual representations into the textual space, and SigLIP (Zhai et al., 2023) as the vision encoder, following empirical validation in (Tong et al., 2024). **To further investigate the generalizability of MMA, we further include Qwen2.5-3B-Instruct (Team, 2024) as the LLM backbone in the main comparisons (Sec. 4.3).** Following (Xue et al., 2024), we set the resolution of pretraining images to 384×384 pixels to align with SigLIP. The number of vision tokens is fixed at 144. The global batch sizes for pretraining and finetuning are 192 and 64, respectively. To analyze the impact of modality-mutual attention, we adopt a short training schedule for rapid iteration across all models. Specifically, we set the number of pretraining and finetuning steps to 50k and 5k, respectively. More details are provided in Appendix C.

## 4.2 THE EFFECTS OF DUAL-ORDER TRAINING

To delve into the effects of incorporating cross-modal interactions between the PT and SFT stages, the dual-order training (DOT) method is included along with three variants to examine the impact of cross-modal interactions on multimodal performance: 1) without T&I order in the PT stage (w/o T&I)$_{PT}$; 2) without I&T order in the PT stage (w/o I&T)$_{PT}$; 3) without T&I order in the SFT stage (w/o T&I)$_{SFT}$; 4) without I&T order in the SFT stage (w/o I&T)$_{SFT}$. As shown in Figure 1, the comparison between (I&T)$_{PT}$ + (I&T)$_{SFT}$ and DOT highlights the importance of considering cross-modal interactions, showing that training the model on the same dataset but incorporating an additional text-then-image order in both the PT and SFT stages improves vision-centric performance (e.g., POPE, MM-Vet, and CV-Bench). This signifies that explicitly enabling modalities to interact to each other enhances vision-centric capabilities, particularly for tasks where images need to be interpreted in relation to object-related text queries. Although removing any order during training generally results in inferior performance compared to DOT, some configurations still yield better results. This detrimental effect indicates that simply introducing cross-modal interactions with different orders **requires a significantly extended training schedule to enforce the understanding of such cross-order behaviors between multimodal samples, as shown in Appendix E.1. Consequently, this requirement poses a challenge by directly correlating the performance gain with a prohibitive increase in computational overhead.** Furthermore, adding the T&I order in the SFT stage ((w/o T&I)$_{PT}$) is more prone to negative impacts than introducing it in the PT stage ((w/o T&I)$_{SFT}$), which demonstrates that the PT stage with diverse input orders is able to help the model learn more robust multimodal representations.

Table 1: Impacts on various text-to-image attention flow. **The best performance in each backbone is in boldface while the second-best result is underlined.**

| | General | | | | | Knowledge | | Vision-Centric | | | | |
|---|---|---|---|---|---|---|---|---|---|---|---|---|
| | $\text{MME}^{\text{P}}$ | $\text{MME}^{\text{C}}$ | MMB | $\text{SEED}^{\text{I}}$ | $\text{LLaVA}^{\text{W}}$ | MMMU | $\text{MathV}^{\text{mini}}$ | POPE | MM-Vet | RealWorldQA | $\text{CV-Bench}^{\text{2D}}$ | $\text{CV-Bench}^{\text{3D}}$ |
| **LLaMA-3.2-3B-Instruct** | | | | | | | | | | | | |
| (w/o T&I)$_{\text{PT}}$ | 1015.7 | 199.8 | 43.8 | 54.1 | 35.9 | 22.2 | 20.3 | 62.3 | 15.4 | 32.7 | 37.9 | 49.5 |
| (w/o I&T)$_{\text{PT}}$ | 1003.4 | 201.5 | 44.0 | 52.5 | 35.2 | 23.1 | 20.1 | 67.4 | 17.9 | 32.4 | 40.8 | 48.6 |
| (w/o T&I)$_{\text{SFT}}$ | 1134.2 | **230.9** | 51.3 | **59.2** | 38.6 | 23.8 | 18.9 | 73.5 | 20.3 | 37.8 | 37.5 | 50.7 |
| (w/o I&T)$_{\text{SFT}}$ | 1128.1 | 217.7 | **51.6** | 58.5 | 34.1 | 23.3 | 21.4 | 72.7 | 18.1 | 35.6 | 39.1 | 51.4 |
| DOT (Ours) | **1219.5** | 223.0 | 46.9 | 55.7 | **43.8** | **23.9** | **22.8** | **77.0** | **22.7** | **42.0** | **46.7** | **52.9** |
| **Phi-3.5-Mini-Instruct** | | | | | | | | | | | | |
| (w/o T&I)$_{\text{PT}}$ | 1046.3 | 226.4 | 31.7 | 45.1 | 38.1 | 27.2 | 23.8 | 65.0 | 17.2 | 40.1 | 53.2 | 54.8 |
| (w/o I&T)$_{\text{PT}}$ | 1013.2 | 208.6 | 32.0 | 43.3 | 37.9 | 27.7 | 22.4 | 70.4 | 20.6 | 39.5 | **55.4** | 53.0 |
| (w/o T&I)$_{\text{SFT}}$ | 1194.8 | **289.3** | **58.5** | **61.1** | 40.2 | 28.0 | 21.9 | 79.0 | 22.8 | 47.8 | 41.4 | **63.0** |
| (w/o I&T)$_{\text{SFT}}$ | 1166.2 | 264.3 | 58.4 | 60.8 | 36.9 | 26.7 | 23.1 | 76.8 | 20.4 | 46.9 | 43.3 | 61.2 |
| DOT (Ours) | **1267.8** | 251.4 | 43.8 | 54.7 | **47.5** | **30.7** | **25.6** | **82.7** | **25.0** | **50.5** | 52.2 | 58.1 |

## 4.3 MAIN COMPARISONS OF MMA

To evaluate the effectiveness of the proposed modality-mutual attention (MMA), we conduct a comprehensive comparison against several approaches under identical training configurations, pretraining, and supervised finetuning datasets to ensure fair, apples-to-apples comparisons: **1) (I&T)$_{\text{PT}}$ + (I&T)$_{\text{SFT}}$** refers to the widely used causal attention pipeline (e.g., Molmo (Deitke et al., 2024), BLIP-3 (Xue et al., 2024), and Cambrian (Tong et al., 2024)), where image tokens are placed before text tokens in both the pretraining and finetuning stages (i.e., steps (b) and (d) in Figure 3). **2) Concentric Causal Attention (CCA)** (Xing et al., 2024) modifies the positions and attention masks of image tokens and represents the state-of-the-art method for mitigating object hallucinations without introducing additional parameters or datasets. **3) Mixed-Attention (MA)** (Xie et al., 2025) processes text tokens with causal attention and image tokens with full attention.

Table 2 summarizes the overall performance across these methods, demonstrating that our MMA approach consistently outperforms all baselines and variants. Quantitatively, MMA achieves performance improvements ranging from 1% to **28%** compared to CCA as well as MA, and the qualitative comparisons are illustrated in Appendix G. Since DOT remains deficient for overall multimodal understanding while doubling the training time, our modality-mutual attention (MMA) consistently outperforms all approaches across nearly all benchmarks. In general, CCA and MA, despite not introducing additional parameters or increasing training time, surpass both the (I&T)$_{\text{PT}}$ + (I&T)$_{\text{SFT}}$ and DOT, highlighting the inefficiencies of the conventional vision-language framework. Meanwhile, our proposed MMA significantly and consistently outperforms CCA and MA across various benchmarks, which is attributed to its unlocked design that enables effective cross-modal attention within the LLM. Notably, compared to the (I&T)$_{\text{PT}}$ + (I&T)$_{\text{SFT}}$, MMA improves not only vision-centric tasks but also general VQA tasks. In contrast, CCA's focus on central image positions negatively impacts performance on the MME benchmark, while MA enabling full attention only across image tokens remains inferior to our MMA.

## 4.4 ANALYSIS ON MMA

**We investigate the inner workings of MMA across general (MME$_{\text{C}}$, MME$_{\text{P}}$), knowledge-intensive (MMMU), and vision-centric (POPE) scenarios, as shown in Figure 6. Specifically, we analyze the average attention distribution from image tokens to text tokens within MMA. We apply NLTK (Bird, 2006) for POS tagging and group text tokens into six categories: Objects**

Table 2: Apples-to-apples comparisons between the commonly-used causal attention $(I\&T)_{PT}$ + $(I\&T)_{SFT}$ pipeline (e.g., BLIP-3 (Xue et al., 2024)), CCA (Xing et al., 2024), MA (Xie et al., 2025), DOT, and our modality-mutual attention (MMA) under the same training configurations and datasets with **3** different LLM backbones. Note that CCA, MA, and MMA also follow the $(I\&T)_{PT}$ + $(I\&T)_{SFT}$ training pipeline. Performance improvements are calculated relative to the best-performing CCA or MA approaches. The best performance in each column is in **boldface** while the second-best result is underlined. Benchmark names are abbreviated due to space limits. The detailed metrics are reported in Appendix F.2.

| | General | | | | | Knowledge | | Vision-Centric | | | | |
|---|---|---|---|---|---|---|---|---|---|---|---|---|
| | $MME^P$ | $MME^C$ | MMB | $SEED^I$ | $LLaVA^W$ | MMMU | $MathV^{mini}$ | POPE | MM-Vet | RealWorldQA | $CV\text{-}Bench^{2D}$ | $CV\text{-}Bench^{3D}$ |
| **LLaMA-3.2-3B-Instruct** | | | | | | | | | | | | |
| $(I\&T)_{PT}$ + $(I\&T)_{SFT}$ | 1179.0 | 215.1 | 57.2 | 60.8 | 41.5 | 23.4 | 20.6 | 74.9 | 21.8 | 41.7 | 40.4 | 49.8 |
| CCA (NeurIPS-24) | 1196.8 | 221.9 | 60.3 | 61.4 | 48.9 | 25.6 | 21.1 | 76.5 | 26.6 | 43.9 | 48.1 | 52.3 |
| MA (ICLR-25) | 1214.0 | 235.6 | 60.7 | 61.0 | 49.0 | 28.5 | 20.8 | 77.3 | 27.9 | 42.8 | 47.2 | 52.9 |
| DOT (Ours) | 1219.5 | 223.0 | 46.9 | 55.7 | 43.8 | 23.9 | **22.8** | 77.0 | 22.7 | 42.0 | 46.7 | 52.9 |
| MMA (Ours) | **1258.2** | **263.8** | **64.6** | **65.0** | **53.7** | **37.1** | 22.8 | **79.2** | **29.1** | **44.7** | **50.4** | **54.1** |
| Improvements | 3.6% | 12.0 % | 6.4% | 5.9% | 9.6% | **28.4**% | 8.1% | 2.5% | 4.3% | 1.8% | 4.8% | 2.7% |
| **Phi-3.5-Mini-Instruct** | | | | | | | | | | | | |
| $(I\&T)_{PT}$ + $(I\&T)_{SFT}$ | 1226.3 | 258.2 | 64.9 | 64.1 | 47.0 | 31.1 | 24.2 | 79.8 | 24.3 | 50.6 | 45.2 | 54.3 |
| CCA (NeurIPS-24) | 1212.7 | 243.6 | 67.4 | 65.3 | 54.0 | 34.6 | 25.6 | 81.9 | 29.0 | **52.7** | 56.0 | 62.8 |
| MA (ICLR-25) | 1271.1 | 280.8 | 68.3 | 65.0 | 52.8 | 35.1 | 21.3 | 82.2 | 29.3 | 51.5 | 54.4 | 59.5 |
| DOT (Ours) | 1267.8 | 251.4 | 43.8 | 54.7 | 47.5 | 30.7 | 25.6 | **82.7** | 25.0 | 50.5 | 52.2 | 58.1 |
| MMA (Ours) | **1363.7** | **315.4** | **71.8** | **67.1** | **59.6** | **39.4** | **26.4** | **82.7** | **30.2** | 52.3 | **57.8** | **64.1** |
| Improvements | 7.3% | 12.3% | 5.1% | 2.8% | 10.4% | **12.3**% | 3.1% | **0.6%** | 3.1% | - | 3.2% | 2.1% |
| **Qwen2.5-3B-Instruct** | | | | | | | | | | | | |
| $(I\&T)_{PT}$ + $(I\&T)_{SFT}$ | 1278.4 | 266.2 | 66.4 | 65.2 | 50.1 | 32.0 | 24.0 | 79.6 | 26.1 | 50.8 | 48.5 | 57.2 |
| CCA (NeurIPS-24) | 1296.7 | 273.1 | 68.7 | 66.4 | 55.3 | 34.5 | 25.3 | 81.4 | 29.6 | 52.9 | 54.7 | 61.5 |
| MA (ICLR-25) | 1324.9 | 293.0 | 70.1 | 66.1 | 54.6 | 36.0 | 23.7 | 82.1 | 30.0 | 52.3 | 53.9 | 62.8 |
| DOT (Ours) | 1315.4 | 269.4 | 46.8 | 66.8 | 48.9 | 32.7 | 26.1 | 82.9 | 26.7 | 51.2 | 51.6 | 61.2 |
| MMA (Ours) | **1408.6** | **327.5** | **73.3** | **68.1** | **60.7** | **40.9** | **27.0** | **84.4** | **31.5** | **54.1** | **57.1** | **65.2** |
| Improvements | 6.3% | 11.8% | 4.6% | 2.6% | 9.8% | **13.6**% | 3.4% | 1.8% | 5.0% | 2.3% | 4.4% | 3.8% |

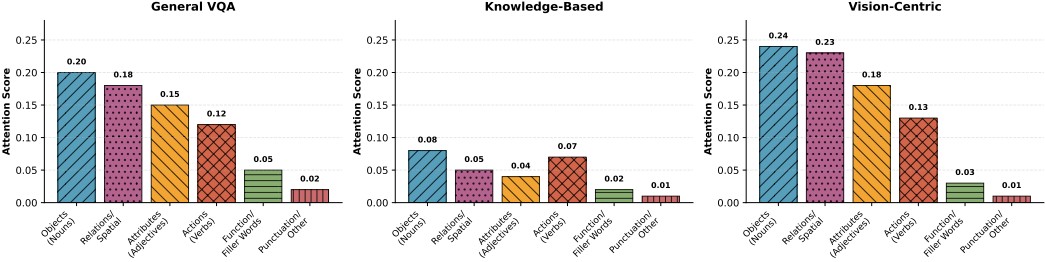

Figure 6: **Attention distributions from image tokens to text tokens, grouped into six linguistic categories derived using NLTK, for MMA under general, knowledge, and vision-centric scenarios.**

**(nouns), Relations/Spatial, Attributes (adjectives), Actions (verbs), Function/Filler Words, and Punctuation/Other.**

**For tasks requiring high-precision visual verification (e.g., hallucination checking), MMA exhibits its most concentrated attention (Nouns at 0.24, Relations at 0.23). This highly fo-**

**cused mechanism selectively grounds the critical entities and relationships, directly explaining our superior factual alignment performance. For knowledge-intensive tasks, the overall cross-modal attention is significantly lower (Nouns peak at 0.08). This indicates that the model correctly recognizes the task's reliance on the LLM's parametric knowledge, prioritizing intra-modal refinement while only establishing the minimal, necessary conceptual link (Nouns/Verbs) with the text. Additionally, the general scenarios demonstrates a balanced attention distribution (Nouns at 0.20) suitable for the diverse challenges of general competence benchmarks. These results confirm that MMA is able to dynamically enable visual tokens to attend to textual tokens based on different tasks.**

### 4.5 CASE STUDY: WHAT INFORMATION ARE IMAGE TOKENS EXTRACTING FROM TEXT TOKENS?

To study the rationale of MMA, we visualize the attention map of MMA from a test sample of RealWorldQA, as shown in Figure 7. We first segment the pixel coordinates for tater tots, hamburger, water, table, and napkins using SAM (Kirillov et al., 2023), and then aggregate the corresponding attention scores from each entity to the query text. The analysis shows that entity pixels (e.g., tater tots, hamburger, water in the y-axis) primarily attend to the text token "where", indicating that unlocking attention flow from image to text allows image tokens to consider the text query describing relations between entities (e.g., locations) for producing precise vision-centric answers. Moreover, the entity pixels of water attend more significantly to the text token "compared", which implies that image tokens learn to link targeted regions (e.g., water) to the goal in the user query (e.g., "compared").

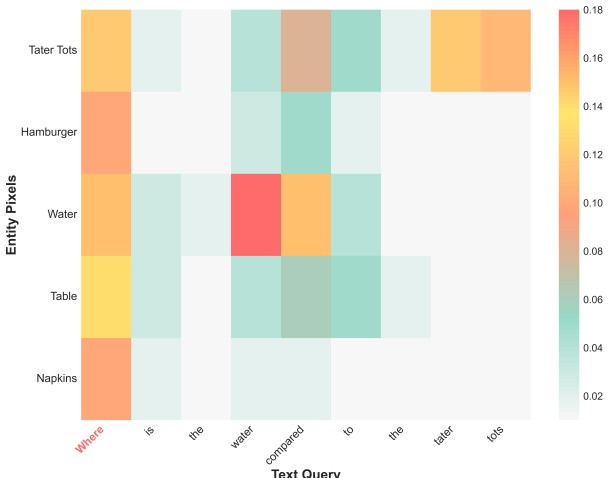

Figure 7: The aggregated attention scores from entity pixels (y-axis) to the text query (x-axis) for a test case from RealWorldQA. The corresponding image is shown in Figure 11.

## 5 CONCLUSIONS

This paper proposes a multimodal LLM with the novel modality-mutual attention (MMA) design to tackle vision-language misalignment from a fundamental architecture perspective. Distinct from existing MLLMs in which image tokens fail to consider the information from the text tokens due to the causal attention design in the LLM, our modality-mutual attention enables tokens from the earlier modality to be able to attend to tokens from the latter modality by unlocking attention masks for cross-modal interactions. Our proposed MMA significantly outperforms the baseline with causal attention, intuitive dual-order training methods, and the state-of-the-art approaches across 3 LLM backbones on 12 extensive multimodal understanding benchmarks without introducing extra parameters. We believe modality-mutual attention serves as a generic and scalable design for other multi-modalities as discussed in Appendix A.

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

## A BROADER IMPACTS, LIMITATION, AND LLM USAGE

### A.1 BROADER IMPACTS

As our modality-mutual attention operates on the attention mask in the LLM, it benefits not only vision-language generative pretraining (e.g., (Fini et al., 2024)) but also generalizes to X-language modalities, where X represents other modalities (e.g., audio). Moreover, this design can be incorporated into other multimodal data (e.g., image-audio) that involve causal attention in generation. It is also expected to be scalable for hierarchical inputs, such as tabular structures. By open-sourcing our code and models without introducing additional parameters, we aim to provide researchers with the tools to tackle real-world multimodal challenges and drive advancements in MLLMs from a fundamental perspective.

### A.2 LIMITATION

Despite its effective multimodal understanding capabilities, the major limitation of AKI is that the proposed modality-mutual attention only provides benefits during the SFT stage. Applying it directly in the PT stage leads to information leakage–earlier text tokens can access future text tokens through the attention path to image tokens, which, in turn, can attend to future text tokens. This issue arises because the PT stage relies solely on captioning samples, which lack specific questions or inquiries as seen in the SFT stage that could explicitly restrict image tokens' attentions. We leave this for future work, as we believe that addressing multimodal misalignment from a fundamental design perspective remains an unexplored yet critical challenge.

### A.3 LLM USAGE

In addition to using LLMs/VLMs for developing and evaluating our proposed methods, we use LLM for polishing the manuscript.

## B COMPARISONS WITH STATE-OF-THE-ART MLLMs

### B.1 IMPLEMENTATION DETAILS OF ❦AKI-4B: MLLMs WITH MODALITY-MUTUAL ATTENTION (MMA)

For our full-scale AKI-4B model, the numbers of pretraining and finetuning steps are enlarged to 275k (250k for Blip3-kale and 25k for Blip3-ocr-200m) and 10k, respectively. Moreover, OCR data is incorporated to enhance the model's ability to understand text within images. We focus on relatively smaller scale compared to existing advancements on large-scale models, which has gained attention due to their cost-effectiveness for deployment on edge devices, since the objective of this paper is to demonstrate improvements driven by fundamental model modifications.

### B.2 COMPARISONS

To push the limits of MLLMs with the MMA design, we evaluate our AKI-4B model, trained on an extended schedule, against state-of-the-art MLLMs, as shown in Table 3. Our results show that AKI-4B outperforms 7B-scale MLLMs with comparable training data across all benchmarks except for $MME_P$, highlighting the potential of integrating MMA into the conventional MLLM pipeline.

Despite being trained on significantly smaller datasets (e.g., 6.7B pretraining tokens in AKI-4B vs. 1.4T pretraining tokens in QWen2-VL), with lower image resolutions (e.g., 384x384 in AKI-4B vs. 2016×2016 in MM-1.5 as well as 1344x1344 in MiniCPM-V2) and fewer vision tokens (e.g., 144 in AKI-4B vs. 640 in MiniCPM-V2), AKI-4B demonstrates highly remarkable performance in terms of general as well as vision-centric tasks, even achieving the best results for $MME_P$ and $LLaVA_W$. Since we incorporate only a small amount of knowledge-based fine-tuning data, AKI-4B is expected to be inferior to existing MLLMs in the knowledge domain. Although our AKI-4B model does not support interleaved inputs, it still performs on par with other MLLMs on the MMMU benchmark, which includes some interleaved samples. For such cases, we use only the first image for inference. These findings suggest that existing MLLMs could further expand the boundaries of multimodal understanding by integrating our proposed MMA design.

Table 3: Comparisons with state-of-the-art models. Results are from the OpenVLM leaderboard, the results with * are from (Xue et al., 2024), and the results with † refer to the corresponding official papers. The best performance within the 3B group is in **boldface**. The detailed metrics for each benchmark of AKI-4B are reported in Appendix F.2.

| | General | | | | | Knowledge | | Vision-Centric | | | | |
|---|---|---|---|---|---|---|---|---|---|---|---|---|
| | $MME^P$ | $MME^C$ | $MMB^{dev}$ | $SEED^I$ | $LLaVA^W$ | MMMU | $MathV^{mini}$ | POPE | MM-Vet | RealWorldQA | $CV\text{-}Bench^{2D}$ | $CV\text{-}Bench^{3D}$ |
| **3B Model Comparisons** | | | | | | | | | | | | |
| *Proprietary* | | | | | | | | | | | | |
| MM1-3B† (McKinzie et al., 2024) | 1482.5 | 279.3 | 67.8 | 68.8 | 72.1 | 33.9 | 32.0 | 87.4 | 43.7 | - | - | - |
| MM1.5-3B† (Zhang et al., 2024) | 1478.4 | 319.6 | - | **72.4** | 73.0 | 37.1 | 44.4 | **88.1** | 41.0 | 56.9 | - | - |
| *Open-source* | | | | | | | | | | | | |
| DeepSeek-VL-1.3B (Lu et al., 2024a) | 1306.6 | 225.0 | - | 66.0 | 51.1 | 33.8 | 30.7 | 85.9 | 29.2 | 49.7 | - | - |
| MiniCPM-V2-3B (Yao et al., 2024) | 1411.4 | 396.8 | 69.1 | 67.1 | 69.2 | 38.2 | 39.8 | 86.3 | 41.0 | 55.8 | - | - |
| VILA-1.5-3B (Lin et al., 2024) | 1379.3 | 268.2 | 62.4* | 68.0 | 65.5 | 34.2 | 31.8 | 86.8 | 38.8 | 53.2 | 50.1* | 60.3* |
| Phi-3-Vision-4B (Abdin et al., 2024) | 1205.1 | 302.9 | 74.2* | 70.9 | 63.9 | 46.1 | 44.8 | 83.7 | 44.1 | 58.8 | 60.7* | 68.2* |
| BLIP-3-4B (Xue et al., 2024) | 1487.6 | 302.9 | 76.0* | 71.8 | 69.8 | 40.1 | 40.1 | 86.9 | 41.0 | 61.6 | **66.2*** | **75.4*** |
| Qwen2-VL-2B (Wang et al., 2024) | 1485.1 | **413.9** | - | **72.4** | 50.5 | **42.2** | **48.0** | 87.3 | **51.5** | 60.7 | - | - |
| 🍁AKI-4B (Ours) | **1491.9** | 362.9 | 73.1 | 69.4 | **74.6** | 38.7 | 32.1 | **88.1** | 40.8 | **62.9** | 62.3 | 71.8 |
| **7B Model Comparisons** | | | | | | | | | | | | |
| LLaVA-1.5-7B (Liu et al., 2024b) | 1506.2 | 302.1 | 64.3† | 65.8 | 61.8 | 35.7 | 25.5 | 86.1 | 32.9 | 54.8 | - | - |
| Honeybee-C-7B† (Cha et al., 2024) | 1584.2 | 307.1 | 70.1 | 64.5 | 67.1 | 35.3 | - | 83.2 | 34.9 | - | - | - |

Table 4: Hyperparameters for pretraining and finetuning experiments in Section 4.3 and for AKI.

| Configurations | Pre-Training | Supervised Finetuning |
|---|---|---|
| Vision Encoder | siglip-so400m-patch14-384 | |
| VL-Connector | Perceiver Resampler | |
| LLM | Phi-3.5-mini-instruct | |
| Trainable Modules | VL-Connector, LLM | |
| # Visual tokens | 144 | |
| Batch Size | 192 | 64 |
| Learning Rate | 1e-4 | 2e-5 |
| Minimum LR | 1e-5 | 1e-6 |
| LR Schedule | Cosine Decay | |
| Warmup Steps | 2000 | 150 |
| Training Steps | 50k (Sec. 4.3)/275k (AKI) | 5k (Sec. 4.3)/10k (AKI) |
| Weight Decay | 1e-2 | 1e-4 |
| Optimizer | AdamW | |
| Gradient Clipping | 1.0 | |
| Precision | amp_bf16 | |
| Max Length | 128 | 512 |

## C  IMPLEMENTATION DETAILS

In this section, we provide detailed settings for our implementations, training configurations, as well as the prompt template.

### C.1  HYPER-PARAMETER SETTINGS

Our implementation is built on PyTorch with Fully Sharded Data Parallel (FSDP), leveraging the OpenFlamingo codebase[3]. To enhance computational efficiency and optimize throughput, we employ PyTorch's Automatic Mixed Precision (AMP) with bfloat16, which allows for reduced mem-

---

[3]https://github.com/mlfoundations/open_flamingo

Table 5: Statistics for supervised finetuning datasets.

| Task | Dataset | Total Size | Sampling Probability |
|------|---------|-----------|---------------------|
| Open-ended VQA | VQAv2 | 83k | 10.3% |
| | GQA | 72k | 10.3% |
| | VSR | 8k | 2.6% |
| | OCRVQA | 80k | 10.3% |
| Multi-choices VQA | A-OKVQA | 50k | 10.3% |
| | ScienceQA | 12k | 10.3% |
| Referring expression | RefCOCO | 120k | 10.3% |
| | RefCOCOg | 80k | 10.3% |
| | RefCOCO+ | 120k | 10.3% |
| | VisualGnome | 86k | 4.7% |
| Instruction-following | LLaVA-150k | 158k | 10.3% |

ory usage while maintaining numerical stability. A comprehensive summary of the hyperparameters used in our experiments is provided in Table 4.

For model training, we conducted experiments on a single node equipped with $8 \times$ A100 (80GB) GPUs. The training duration varied depending on the model configuration: Each model in Section 4.3 required approximately 2 days for pretraining and 1 day for supervised finetuning, while DOT requires around 4 days for pretraining and 2 days for supervised finetuning. The AKI-4B model, given its larger scale, required around 3 weeks for pretraining and 2 days for supervised finetuning. For all benchmark evaluations, we employed greedy decoding to generate responses, ensuring consistency across different models and datasets.

For supervised finetuning, we adopt a sampling ratio for the collected dataset similar to (Cha et al., 2024), as detailed in Table 5. To assess the impact of different sampling strategies, we empirically compared this hand-crafted ratio with a random ratio, where each data sample is selected based on a uniform distribution. Our findings indicate that using the hand-crafted ratio results in slightly better performance compared to the random ratio, suggesting that a carefully balanced dataset composition contributes to improved multimodal learning. Specifically, VQAv2, GQA, OCRVQA, A-OKVQA, ScienceQA, RefCOCO, RefCOCOg, RefCOCO+, and LLaVA-150k each have an equal sampling ratio of 10.3%. VSR is assigned a lower sampling ratio of 2.6%, reflecting its unique task characteristics. VisualGnome is set at 4.7%, as it contains most samples.

## C.2 PROMPT TEMPLATES

To ensure the reproducibility of our model, we provide detailed dataset templates for both the pretraining and supervised finetuning stages in Table 6. For pretraining, we follow the template format introduced in (Awadalla et al., 2023), structured as:

$$\texttt{<image>\{image\_tokens\}\{caption\}<|endofchunk|>}, \qquad (7)$$

where image tokens are followed by the caption, and <|endofchunk|> is used as a boundary marker.

For supervised finetuning, we adopt the dataset templates from (Cha et al., 2024), with the exception of the referring expression tasks (RefCOCO, RefCOCOg, RefCOCO+, and VisualGnome). Instead, we employ the template with "Provide a short description for this region." following (Liu et al., 2023b) based on our empirical experiments. Additionally, we do not apply de-duplication techniques in data processing. Multi-turn dialogues are converted into multiple single-turn samples, ensuring consistency across different datasets.

## D DATASET DETAILS

In this section, we provide a brief overview of the key aspects for each dataset we use for pretraining and supervised finetuning.

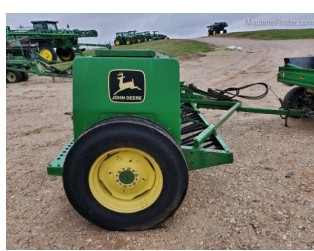 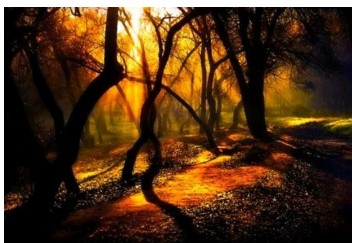 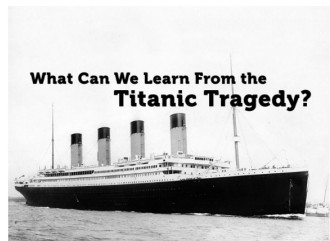

In the provided image, a John Deere tractor attachment from 1991 is depicted. The attachment features a large wheel in the foreground, encircled by a yellow rim. The logo of a leaping deer in black and white adorns the center. Surrounding the attachment are other green tractors and machinery, while the ground consists of a mixture of gravel and dirt. The overcast sky indicates a cloudy day.

In the enchanting forest of Moldova, depicted in the image, the sun's rays filter through the dense canopy during either dawn or dusk, casting a warm, golden hue over the landscape. The forest floor is adorned with a rich layer of fallen leaves, and a narrow path weaves through the scene, inviting exploration. The ancient trees, with their twisted and gnarled trunks, testify to the forest's enduring history, adding to the overall ambiance of tranquility and mystery.

The RMS Titanic, a renowned ocean liner that met its tragic end in 1912, is depicted in this black-and-white photograph. With four large smokestacks gracing its deck, the ship is shown amidst the water's expanse, displaying an apparent motion that stirs the surrounding waves. Reflecting on this maritime disaster, we ask: What Can We Learn From the Titanic Tragedy?

Figure 8: Illustrations sampled from the Blip3-kale dataset.

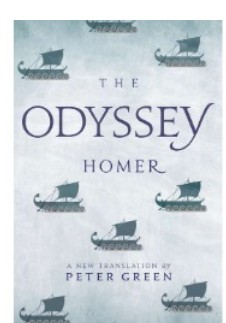 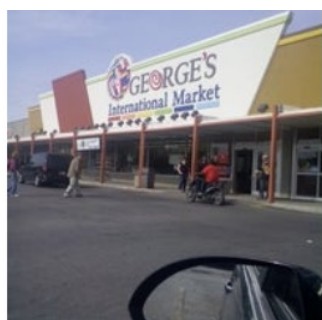 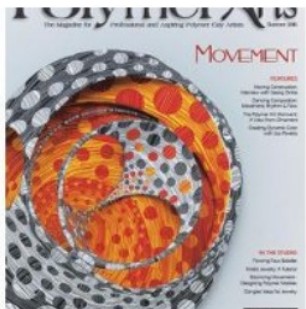

The image contains the text \"THE\", the text \"ODYSSEY\", the text \"HOMER\", the text \"A NEW TRANSLATION BY\", the text \"PETERGREEN\"

The image contains the text \"GEORGE'S\"

The image contains the text \"MOVEMENI\"

Figure 9: Illustrations sampled from the Blip3-OCR dataset.

Table 6: Detailed prompt template for each dataset. The templates are mainly followed (Cha et al., 2024; Liu et al., 2023b). {system message} in each dataset template is replaced with the system message. {image_tokens}, {questions}, and {answer} are replaced based on each data sample. The bounding box coordinates {bbox} are normalized as $[x_{min}, y_{min}, x_{max}, y_{max}]$.

| Task | Dataset | Prompt Template |
|------|---------|-----------------|
| **Pre-Training Template** | | |
| Captioning | BLIP3-KALE | `<image>`{image_tokens}{caption}`<|endofchunk|>` |
| **Supervised Finetuning Template** | | |

**System Message:**
`<|system|>`
A chat between a curious user and an artificial intelligence assistant.
The assistant gives helpful, detailed, and polite answers to the user's questions. `<|end|>`

| Task | Dataset | Prompt Template |
|------|---------|-----------------|
| Open-ended VQA | VQAv2 GQA VSR OCRVQA | {system message} `<|user|>` `<image>` {image_tokens} Answer the question using a single word or phrase. {question}`<|end|>` `<|assistant|>` {answer} |
| Multi-choices VQA | A-OKVQA ScienceQA | {system message} `<|user|>` `<image>` {image_tokens} Answer with the option's letter from the given choices directly. {question} Options: {options} `<|end|>` `<|assistant|>` {answer} |
| Referring expression | RefCOCO RefCOCOg RefCOCO+ VisualGnome | {system message} `<|user|>` `<image>` {image_tokens} Provide a short description for this region. `<bbox>`{bbox}`</bbox>` `<|end|>` `<|assistant|>` {answer} |
| Instruction-following | LLaVA-150k | {system message} `<|user|>` `<image>` {image_tokens} {question} `<|end|>` `<|assistant|>` {answer} |

## D.1 PRETRAINING DATA

We adopt Blip3-kale (Awadalla et al., 2024) and Blip3-ocr-200m (Xue et al., 2024) as the pretraining datasets. Blip3-kale consists of 218M image-text captions by augmenting synthetic dense image captions with web-scale to generate factually grounded image captions, where the images are originally come from the Datacomp-1B dataset (Gadre et al., 2023). The Blip3-OCR-200M dataset comprises 12 hierarchical levels, ranging from coarse to fine-grained representations for text-rich images. We utilize Level 1 (basic text extraction) to enhance the model's ability to recognize and interpret text within images. For Blip3-Kale, approximately 10 million and 100 million captioning pairs were sampled for the experiments in Section 4.3 and the final AKI-4B model, respectively. Additionally, around 5 million samples from Blip3-OCR-200M were used to train the AKI-4B model. Some examples sampled from the Blip3-kale and Blip3-ocr-200m datasets are provided in Figures 8 and 9, respectively.

## D.2 SUPERVISED FINETUNING DATA

Generally, the aim of supervised finetuning is to enable the model to learn specific skills from related domains (e.g., VQA). Since the main focus in this paper is to improve multimodal performance from the fundamental architecture perspective, instead of from scaling numbers of datasets and parameters like Cambrian and QWen, we mainly use VQA and referring expression datasets as our visual instruction tuning datasets. Specifically, the brief introduction of each dataset is describe as follows:

- **VQAv2 (Goyal et al., 2017):** A large-scale Visual Question Answering dataset containing images paired with open-ended questions and human-annotated answers from various capabilities, e.g., visual grounding, spatial understnading, visual recognition.

- **GQA (Hudson & Manning, 2019):** is a scene understanding dataset featuring compositional questions over real-world images, including semantic representations of the scenes and questions to mitigate language priors and conditional biases for different question types.

- **VSR (Liu et al., 2023a):** is a visual spatial reasoning dataset containing text-image pairs with various types of spatial relations in English (e.g., under, in front of), which is an important criteria for multimodal understanding.

- **OCRVQA (Mishra et al., 2019):** learns the ability of the VQA task that interprets text in images, which is often known as the OCR capability.

- **A-OKVQA (Schwenk et al., 2022):** aspires to provide the requirement of understanding form of commonsense reasoning about the image. The questions in A-OKVQA generally cannot be answered by simply querying a knowledge base.

- **ScienceQA (Lu et al., 2022):** consists of image-text multiple choice questions with diverse science topics, including annotations of their answers with corresponding lectures and explanations.

- **RefCOCO (Kazemzadeh et al., 2014):** A dataset for referring expression comprehension, where a model must localize a specific object in an image given a natural language description, collected from interactive games.

- **RefCOCOg (Mao et al., 2016):** A variation of RefCOCO with longer and more descriptive referring expressions, supporting more complex reasoning.

- **RefCOCO+ (Yu et al., 2016):** A referring expression comprehension dataset like RefCOCO but focusing on descriptions without absolute location words, making localization more challenging.

- **VisualGnome (Krishna et al., 2017):** contains densely annotated region descriptions, VQA, object instances, attributes, and relationships, aiming to connect structured image concepts to language.

- **LLaVA-150k (Liu et al., 2023b):** is a GPT-generated dataset for learning multimodal instruction-following capabilities. Its aim is to enable the model towards the multimodal capability of GPT-4.

# E  ADDITIONAL EXPERIMENTS

## E.1  ABLATION ON LONGER TRAINING SCHEDULE OF DOT

To further investigate the impacts of DOT, we conducted an ablation by extending the supervised finetuning schedule from 5k to 10k steps (DOT-2x), as shown in Table 7. While DOT-2x achieves superior performance across most benchmarks, indicating that a longer schedule is indeed beneficial for learning cross-modal relations, this improvement comes at the expense of a significant computational cost.

## E.2  RELATIONS BETWEEN MASK SPARSITY AND PERFORMANCE

To delve into the impacts of changing the accessible proportion from image tokens to text tokens, we conducted additional analysis by restricting with 0%, 10%, 30%, 50%, 100%. Here, 0% is exactly the conventional causal attention, and 100% refers to the MMA design. Figure 10 presents the performance curve, which shows monotonic improvements as the accessible to text token ratio increases. This indicates that exposing image tokens to more text context consistently helps the model interpret queries more accurately.

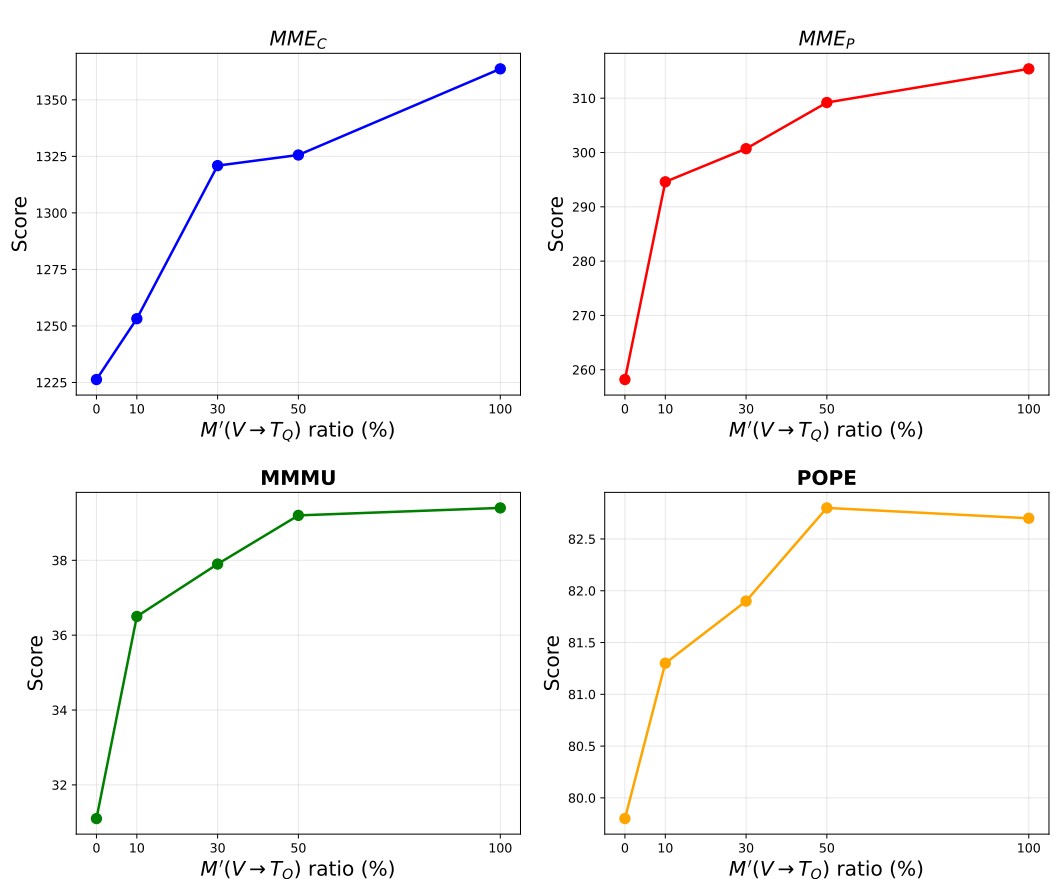

Figure 10: **Performance on MMB, MMMU, and POPE in terms of different accessible proportion.**

Table 7: **Ablation study on longer training schedule of DOT. 2x refers to supervised finetuning DOT with 2x steps.**

| | General | | | | | Knowledge | | Vision-Centric | | | | |
|---|---|---|---|---|---|---|---|---|---|---|---|---|
| | $\text{MME}^{P}$ | $\text{MME}^{C}$ | MMB | $\text{SEED}^{I}$ | $\text{LLaVA}^{W}$ | MMMU | $\text{MathV}^{mini}$ | POPE | MM-Vet | RealWorldQA | $\text{CV-Bench}^{2D}$ | $\text{CV-Bench}^{3D}$ |
| **LLaMA-3.2-3B-Instruct** | | | | | | | | | | | | |
| DOT | 1219.5 | 223.0 | 46.9 | 55.7 | **43.8** | 23.9 | **22.8** | 77.0 | 22.7 | 42.0 | 46.7 | 52.9 |
| **DOT-2x** | **1236.1** | **226.4** | **55.8** | **62.3** | 42.5 | **24.2** | 19.7 | **79.5** | **24.3** | **42.8** | **47.7** | **54.0** |
| **Phi-3.5-Mini-Instruct** | | | | | | | | | | | | |
| DOT | 1267.8 | 251.4 | 43.8 | 54.7 | 47.5 | 30.7 | **25.6** | 82.7 | 25.0 | **50.5** | 52.2 | 58.1 |
| **DOT-2x** | **1311.9** | **276.5** | **54.0** | **64.6** | **51.2** | **36.6** | 25.3 | **84.4** | **29.9** | **50.5** | **52.6** | **61.4** |
| **Qwen2.5-3B-Instruct** | | | | | | | | | | | | |
| **DOT** | 1315.4 | 269.4 | 46.8 | 66.8 | 48.9 | 32.7 | 26.1 | **82.9** | **26.7** | 51.2 | 51.6 | 61.2 |
| **DOT-2x** | **1359.3** | **290.5** | **58.4** | **66.9** | **55.0** | **36.7** | **26.4** | 82.5 | 25.9 | **53.5** | **53.8** | **62.1** |

Table 8: Detailed prompt template for evaluating each benchmark. We omit the system message and special tokens for better readability as in Table 6. The hint in MMB and $\text{MathV}^{mini}$ is included only if provided.

| Benchmark | Prompt Template |
|---|---|
| $\text{MME}^{P|C}$ | Answer the question using a single word or phrase. {question} Please answer yes or no. |
| $\text{MMB}^{dev}$ | Answer with the option's letter from the given choices directly. Hint: {hint}\n Question: {question}\n Options: {options} |
| $\text{SEED}^{I}$ | Answer with the option's letter from the given choices directly. {question}\n Options: {options} |
| $\text{LLaVA}^{W}$ | {question} |
| MMMU | Answer with the option's letter from the given choices directly. {question}\n Options: {options} |
| $\text{MathV}^{mini}$ | Hint: {hint}\n Question: {question} |
| POPE | Answer the question using a single word or phrase. {question} Please answer yes or no. |
| MM-Vet | {question} |
| RealWorldQA | Answer with the option's letter from the given choices directly. {question}\n Options: {options} |
| $\text{CV-Bench}^{2D|3D}$ | Answer with the option's letter from the given choices directly. {question}\n Options: {options} |

# F    BENCHMARK DETAILS

In this section, we provide detailed descriptions of the evaluations conducted in the experiments, including the prompt templates as well as comprehensive metric results for each benchmark.

## F.1    PROMPT TEMPLATE FOR BENCHMARK EVALUATIONS

Table 8 presents the detailed prompt templates used for evaluating each multimodal benchmark to ensure reproducibility. For clarity, we omit the system messages and special tokens, following the format in Table 6. Furthermore, for benchmarks such as MMB and $\text{MathV}^{mini}$, hints are included only when explicitly provided in the sample.

## F.2    DETAILED BENCHMARK SCORES

Beyond demonstrating the effectiveness of our approach, we provide detailed metrics for each evaluation benchmark from the best-performing *Phi-3.5-Mini-Instruct* to offer deeper insights into both

Table 9: Detailed perception and cognition scores for MME, with a maximum score of 200 for each category (row).

| Model | (I&T)$_{PT}$ + (I&T)$_{SFT}$ | CCA | MA | DOT (Ours) | MMA (Ours) | 🍁AKI-4B |
|---|---|---|---|---|---|---|
| **Perception** | | | | | | |
| OCR | 57.5 | 70 | 66 | 65 | 80 | 132.5 |
| Artwork | 133.5 | 131 | 139 | 140 | 135 | 126.8 |
| Celebrity | 109.1 | 72.4 | 102.5 | 111.5 | 122.9 | 124.1 |
| Color | 138.3 | 145 | 153 | 150 | 163.3 | 163.3 |
| Count | 98.3 | 130 | 135 | 133.3 | 148.3 | 175 |
| Existence | 145 | 165 | 165 | 165 | 175 | 195 |
| Landmark | 140.5 | 149.5 | 151.3 | 143.3 | 149.8 | 154.3 |
| Position | 96.7 | 81.7 | 77.6 | 81.7 | 90 | 111.7 |
| Posters | 145.6 | 119.4 | 122.7 | 120.1 | 123.8 | 151 |
| Scene | 161.8 | 148.5 | 159 | 158 | 156.8 | 158.3 |
| Sum | 1226.3 | 1212.7 | 1271.1 | 1267.8 | 1344.9 | 1491.9 |
| **Cognition** | | | | | | |
| Code reasoning | 50 | 62.5 | 53 | 50 | 52.5 | 55 |
| Commonsense reasoning | 105.7 | 93.6 | 105.3 | 101.4 | 117.9 | 117.9 |
| Numerical calculation | 40 | 40 | 52.5 | 42.5 | 52.5 | 50 |
| Text translation | 62.5 | 47.5 | 70 | 57.5 | 92.5 | 140 |
| Sum | 258.2 | 243.6 | 280.8 | 251.4 | 315.4 | 362.9 |

Table 10: Detailed scores (%) for MMB. Abbreviations: AR – Attribute Reasoning, CP – Coarse Perception, FP-C – Fine-grained Perception (cross-instance), FP-S – Fine-grained Perception (single-instance), LR – Logical Reasoning, RR – Relation Reasoning.

| Model | AR | CP | FP-C | FP-S | LR | RR | Overall |
|---|---|---|---|---|---|---|---|
| (I&T)$_{PT}$ + (I&T)$_{SFT}$ | 75.6 | 71.0 | 53.1 | 70.2 | 34.7 | 66.7 | 64.9 |
| CCA | 78.0 | 73.2 | 59.2 | 73.0 | 34.7 | 67.8 | 67.4 |
| MA | 76.2 | 69.8 | 60.1 | 72.5 | 27.8 | 65.6 | 68.3 |
| DOT (Ours) | 56.1 | 53.9 | 27.4 | 46.0 | 21.0 | 40.8 | 43.8 |
| MMA (Ours) | 75.9 | 78.0 | 63.6 | 71.7 | 48.3 | 67.8 | 70.3 |
| 🍁AKI-4B | 76.8 | 81.1 | 66.0 | 75.7 | 46.9 | **65.0** | 73.1 |

the strengths and limitations across different categories from Tables 9 to 17. We believe that sharing comprehensive results can contribute to advancing multimodal understanding. For benchmarks requiring LLM-based evaluation (e.g., MM-Vet), we use GPT-4-0613 as the judgment grader. Additionally, certain benchmarks are excluded from the detailed breakdown since they rely on a single metric (e.g., RealWorldQA).

## G  QUALITATIVE RESULTS

In addition to the quantitative results, we further analyze the qualitative outputs generated by the baselines ((I&T)$_{PT}$ + (I&T)$_{SFT}$) and CCA, as well as our proposed MMA method. Figures 11, 12, 13, and 14 present samples from various multimodal benchmarks, including multiple-choice and open-ended questions. The comparisons between the baseline and CCA reveal that while CCA reduces hallucinations in the multiple-choice case (Figure 11), it still generates responses containing hallucinated objects (e.g., both Figures 12 and 14). Moreover, CCA produces an unrelated response in text-rich understanding tasks (e.g., Figure 13), whereas the baseline at least identifies the correct direction, albeit with incorrect details. In contrast, our MMA method effectively mitigates these issues across both multiple-choice and open-ended questions, demonstrating its ability to enhance response accuracy and coherence.

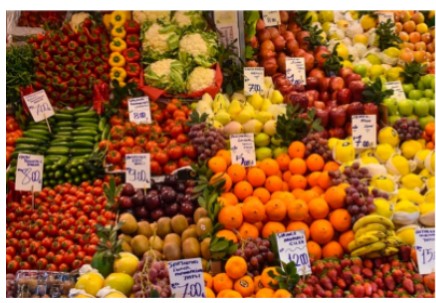

**User:** Where is the water compared to the tater tots?
A. The water is behind to the right of the tater tots.
B. The water is in front to the right of the tater tots.
C. The water is behind to the left of the tater tots.

**(I&T)_PT + (I&T)_SFT:** B

**CCA:** A

**MMA (Ours):** A

Figure 11: Qualitative comparisons among the conventional training pipeline ((I&T)_PT + (I&T)_SFT), CCA, and our MMA method. The sample is taken from RealWorldQA (xAI, 2024).

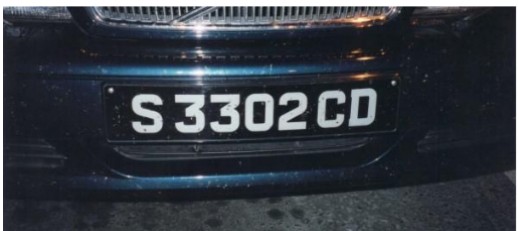

**User:** What fruit is the the right of plums?

**(I&T)_PT + (I&T)_SFT:** apple

**CCA:** strawberries

**MMA (Ours):** oranges

Figure 12: Qualitative comparisons among the conventional training pipeline ((I&T)_PT + (I&T)_SFT), CCA, and our MMA method. The sample is taken from MM-Vet (Yu et al., 2024).

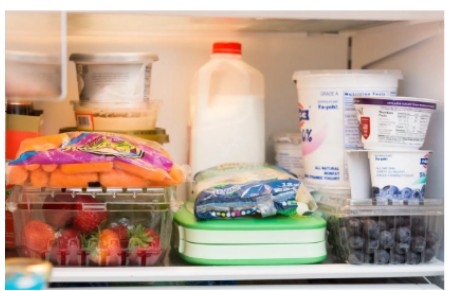

**User:** What are all the scene text in the image?

**(I&T)_PT + (I&T)_SFT:** s 3002 cdd

**CCA:** all the scene text in the image are the license plate

**MMA (Ours):** s3302cd

Figure 13: Qualitative comparisons among the conventional training pipeline ((I&T)_PT + (I&T)_SFT), CCA, and our MMA method. The sample is taken from MM-Vet (Yu et al., 2024).

**User:** What is the brand of the yogurt flavored with blueberry?

**(I&T)_PT + (I&T)_SFT:** The brand of the yogurt is yogurt.

**CCA:** The brand of the yogurt is tropicana.

**MMA (Ours):** The brand of the yogurt is Fage.

Figure 14: Qualitative comparisons among the conventional training pipeline ((I&T)_PT + (I&T)_SFT), CCA, and our MMA method. The sample is taken from LLaVA^W (Liu et al., 2023b).

Table 11: Detailed accuracy for each category in SEED[I].

| Model | (I&T)$_{PT}$ + (I&T)$_{SFT}$ | CCA | MA | DOT (Ours) | MMA (Ours) | AKI-4B |
|---|---|---|---|---|---|---|
| Instance attributes | 64.1 | 66.6 | 67.3 | 53.7 | 69.5 | 69.5 |
| Instance identity | 68.5 | 69.9 | 70.1 | 58.4 | 70.6 | 69.2 |
| Instance interaction | 64.9 | 61.9 | 63.5 | 59.8 | 72.2 | 64.1 |
| Instance location | 55.7 | 55.2 | 56.0 | 45.7 | 56.4 | 55.7 |
| Instance counting | 56.7 | 56.3 | 57.6 | 45.1 | 59.4 | 58.7 |
| Scene understanding | 72.9 | 73.9 | 70.3 | 66.9 | 74.6 | 76.8 |
| Spatial relation | 46.7 | 50.2 | 48.4 | 39.9 | 48.9 | 47.2 |
| Text understanding | 36.9 | 35.7 | 36.5 | 25.0 | 31.0 | 34.6 |
| Visual reasoning | 77.0 | 76.1 | 74.6 | 66.5 | 75.5 | 75.9 |
| Overall | 64.1 | 65.3 | 65.0 | 54.7 | 67.1 | 69.4 |

Table 12: Detailed scores for LLaVA[W].

| Model | Conv | Detail | Complex | Total |
|---|---|---|---|---|
| (I&T)$_{PT}$ + (I&T)$_{SFT}$ | 24.1 | 59.7 | 55.0 | 47.0 |
| CCA | 33.3 | 60.7 | 63.6 | 54.0 |
| MA | 35.7 | 58.5 | 62.9 | 52.8 |
| DOT (Ours) | 29.8 | 53.0 | 58.5 | 47.5 |
| MMA (Ours) | 38.6 | 61.9 | 71.7 | 59.6 |
| AKI-4B | 91.1 | 57.4 | 73.4 | 74.6 |

Table 13: Detailed accuracy (%) of each category for MMMU.

| Model | (I&T)PT + (I&T)SFT | CCA | MA | DOT (Ours) | MMA (Ours) | 🍁AKI-4B |
|---|---|---|---|---|---|---|
| Accounting | 60 | 27 | 50 | 20 | 40 | 40 |
| Agriculture | 20 | 40 | 10 | 0 | 20 | 0 |
| Architecture & Engineering | 0 | 40 | 20 | 20 | **40** | 0 |
| Art | 40 | 50 | 40 | 20 | 40 | 20 |
| Art theory | 40 | 57 | 30 | 20 | 20 | 40 |
| Basic medical science | 60 | 37 | 65 | 80 | 60 | 40 |
| Biology | 60 | 27 | 65 | 20 | 60 | 60 |
| Chemistry | 40 | 13 | 30 | 20 | 20 | 40 |
| Clinical medicine | 40 | 47 | 35 | 40 | 0 | 20 |
| Computer science | 40 | 30 | 30 | 20 | 40 | 40 |
| Design | 40 | 43 | 38 | 40 | 40 | 40 |
| Diagnostics & Laboratory medicine | 40 | 23 | 20 | 20 | **20** | 40 |
| Economics | 40 | 47 | 50 | 40 | 80 | 60 |
| Electronics | 20 | 40 | 30 | 20 | 20 | 40 |
| Energy & Power | 20 | 30 | 40 | 40 | 60 | 60 |
| Finance | 0 | 20 | 20 | 0 | 40 | 0 |
| Geography | 0 | 40 | 20 | 20 | 20 | 20 |
| History | 40 | 37 | 30 | 60 | 60 | 60 |
| Literature | 60 | 70 | 70 | 40 | 60 | 60 |
| Manage | 40 | 27 | 30 | 20 | 60 | 40 |
| Marketing | 40 | 23 | 40 | 20 | 40 | 80 |
| Materials | 40 | 20 | 40 | 40 | 20 | 0 |
| Math | 20 | 33 | 30 | 40 | 40 | 20 |
| Mechanical engineering | 0 | 30 | 10 | 0 | **20** | 40 |
| Music | 20 | 43 | 20 | 40 | 20 | 40 |
| Pharmacy | 20 | 40 | 40 | 60 | 40 | 40 |
| Physics | 20 | 20 | 60 | 20 | 80 | 60 |
| Psychology | 20 | 23 | 20 | 40 | 60 | 40 |
| Public health | 20 | 23 | 20 | 20 | 20 | 60 |
| Sociology | 60 | 43 | 40 | 80 | 60 | 60 |
| Art & Design | 35 | 48 | 40 | 30 | 30 | 35 |
| Business | 36 | 29 | 34 | 20 | 52 | 44 |
| Health & Medicine | 36 | 34 | 42 | 44 | 24 | 40 |
| Humanities & Social science | 45 | 43 | 48 | 55 | 60 | 55 |
| Science | 28 | 27 | 30 | 24 | 44 | 40 |
| Tech & Engineering | 20 | 32 | 27 | 20 | 23 | 26 |
| Overall | 31 | 35 | 35 | 31 | **39** | 39 |

Table 14: Detailed scores for different skills in MathV$^{mini}$.

| | (I&T)PT + (I&T)SFT | CCA | MA | DOT (Ours) | MMA (Ours) | 🍁AKI-4B |
|---|---|---|---|---|---|---|
| Scientific reasoning | 34.4 | 38.5 | 35.7 | 36.9 | 37.7 | 38.5 |
| Textbook QA | 34.8 | 38.6 | 33.3 | 38.6 | 36.1 | 44.3 |
| Numeric commonsense | 19.4 | 20.1 | 20.3 | 16.7 | 20.8 | 27.8 |
| Arithmetic reasoning | 21.3 | 27.2 | 26.9 | 24.9 | 29.8 | 27.5 |
| VQA | 21.2 | 33.0 | 31.8 | 30.2 | 36.9 | 42.5 |
| Geometry reasoning | 19.7 | 16.3 | 13.4 | 20.5 | 18.0 | 26.4 |
| Algebraic reasoning | 24.2 | 20.6 | 19.8 | 23.1 | 20.6 | 24.9 |
| Geometry problem solving | 21.2 | 16.8 | 22.3 | 21.1 | 18.8 | 23.6 |
| Math word problem | 12.9 | 20.4 | 17.9 | 18.3 | 21.5 | 15.6 |
| Logical reasoning | 10.8 | 16.2 | 18.5 | 21.6 | 16.2 | 10.8 |
| Figure QA | 23.4 | 23.4 | 21.1 | 23.4 | 23.1 | 28.6 |
| Statistical reasoning | 22.3 | 19.6 | 23.2 | 23.6 | 22.9 | 27.9 |
| Total | 24.2 | 25.6 | 21.3 | 25.6 | 26.4 | 32.1 |

Table 15: Accuracy, precision, and recall on the popular, adversarial, and random splits for POPE.

| Model | Split | Accuracy | Precision | Recall | Overall |
|---|---|---|---|---|---|
| (I&T)$_{PT}$ + (I&T)$_{SFT}$ | Popular | 82.3 | 92.6 | 70.1 | 79.8 |
| | Adversarial | 80.7 | 88.9 | 70.1 | 78.4 |
| | Random | 83.8 | 96.4 | 70.1 | 81.2 |
| | Overall | 82.2 | 92.6 | 70.1 | 79.8 |
| CCA | Popular | 84.5 | 95.8 | 72.1 | 82.3 |
| | Adversarial | 82.9 | 92.0 | 72.1 | 80.8 |
| | Random | 84.9 | 97.0 | 72.1 | 82.7 |
| | Overall | 84.1 | 94.9 | 72.1 | 81.9 |
| MA | Popular | 83.6 | 96.3 | 72.6 | 83.1 |
| | Adversarial | 83.0 | 90.7 | 72.6 | 80.2 |
| | Random | 82.7 | 96.5 | 72.6 | 83.3 |
| | Overall | 83.1 | 94.8 | 72.6 | 82.2 |
| DOT (Ours) | Popular | 82.5 | 84.1 | 74.6 | 82.5 |
| | Adversarial | 82.9 | 89.4 | 74.6 | 81.4 |
| | Random | 86.0 | 96.6 | 74.6 | 84.2 |
| | Overall | 84.3 | 92.7 | 74.6 | 82.7 |
| MMA (Ours) | Popular | 84.7 | 95.2 | 73.1 | 82.7 |
| | Adversarial | 83.7 | 92.7 | 73.1 | 81.7 |
| | Random | 85.7 | 97.8 | 73.1 | 83.7 |
| | Overall | 84.7 | 95.2 | 73.1 | 82.7 |
| 🍁AKI-4B | Popular | 87.4 | 90.8 | 83.3 | 86.9 |
| | Adversarial | 84.8 | 85.9 | 83.3 | 84.6 |
| | Random | 90.0 | 96.2 | 83.3 | 89.3 |
| | Overall | 87.4 | 90.8 | 83.3 | 86.9 |

Table 16: Detailed scores across different categories for MM-Vet.

| Model | rec | ocr | know | gen | spat | math | Overall |
|---|---|---|---|---|---|---|---|
| (I&T)$_{PT}$ + (I&T)$_{SFT}$ | 36.4 | 17.5 | 11.4 | 10.7 | 18.0 | 0.0 | 24.3 |
| CCA | 40.5 | 19.3 | 18.2 | 14.7 | 26.0 | 3.9 | 29.0 |
| MA | 35.3 | 21.2 | 19.7 | 17.1 | 26.4 | 4.6 | 29.3 |
| DOT (Ours) | 34.6 | 17.8 | 9.9 | 5.6 | 27.6 | 9.6 | 25.0 |
| MMA (Ours) | 39.8 | 22.1 | 20.5 | 16.6 | 27.5 | 3.8 | 30.1 |
| 🍁AKI-4B | 48.2 | 36.1 | 22.6 | 21.4 | 36.1 | 23.1 | 40.8 |

Table 17: Accuracy of each source for CV-Bench.

| Model | ADE20k | COCO | Omni3D | 2D Acc. | 3D Acc. |
|---|---|---|---|---|---|
| (I&T)$_{PT}$ + (I&T)$_{SFT}$ | 41.2 | 49.2 | 54.3 | 45.2 | 54.3 |
| CCA | 48.0 | 64.0 | 62.8 | 56.0 | 62.8 |
| MA | 50.8 | 58.0 | 59.5 | 54.4 | 59.5 |
| DOT (Ours) | 47.2 | 57.1 | 58.1 | 52.2 | 58.1 |
| MMA (Ours) | 52.9 | 62.7 | 64.1 | 57.8 | 64.1 |
| 🍁AKI-4B | 60.6 | 63.6 | 71.8 | 62.1 | 71.8 |

