# OpenReview forum: "Seeing is Understanding: Unlocking Causal Attention into Modality-Mutual Attention for Multimodal LLMs"
_ICLR.cc/2026/Conference — Submitted to ICLR 2026_

### Official Review · Reviewer_owr3 · 2025-10-23

**Soundness:** 2
**Presentation:** 2
**Contribution:** 2
**Rating:** 4
**Confidence:** 4

**Summary:**

This paper focuses on solving the limitations that the earlier modalities (e.g., images) to incorporate information from the latter modalities (e.g., text).  To address this problem, this paper proposes modality-mutual attention to enable image tokens to attend to text tokens. Experiments have shown that MMA achieves state-of-the-art performance in 12 multimodal understanding benchmarks.

**Strengths:**

1. This paper is well motivated.
2. The proposed DOT method provides a plug-and-play advantage.
3. The proposed MMA achieves significant improvement on multiple benchmarks.

**Weaknesses:**

1. The paper needs improvement in quality. For instance, clarify the significance of boldface and underline in Tab.1, which appears to differ from their meanings in Tab.2. There are repeated occurrences of underlines in Tab.2 with non-matching scores.
2. As mentioned in the paper, DOT requires introducing doubled training overhead, as highlighted as a drawback.
3. The novelty of the proposed method in this paper is limited.

**Questions:**

1.  Incorrect Improvements Result: In Tab.2, based on Phi-3.5-Mini-Instruct, MMA achieves 82.7 in POPE. The performance improvement should be (82.7-82.2)/82.2=0.6% instead of 1%.
2. It would be more convincing to validate the proposed method based on other widely used MLLM models (e.g., Qwen2.5VL, LLaVA-OneVision, and InternVL3).
3. In Tab.3, on nearly half of the benchmarks, AKI-4B's performance is worse than Qwen2-VL-2B's. Authors are requested to provide a reasonable explanation.

---

> ### Author Response · Authors · 2025-11-26
> **Response to Reviewer owr3 [1/2]**
>
> We thank the reviewer for the constructive comments and feedback on our paper to improve our paper. Our point-to-point responses to your comments are summarized below.
>
> > **[W1] Different meanings of boldface and underline in Table 1**
>
> Sorry for the confusion. We have revised Table 1 to meet the same meaning as Table 2. Thank you for pointing this out.
>
> > **[W2] DOT requires introducing doubled training overhead, as highlighted as a drawback**
>
> DOT is introduced in our paper as a *conceptually straightforward* baseline for addressing the core problem (Sec. 3.2), and `Reviewer xSEK` also recognized its intuition. Precisely because DOT incurs roughly **2x training overhead**, we were motivated to develop modality-mutual attention (MMA), which achieves the same goal in a significantly more effective and efficient manner (as shown in Table 2).
>
> > **[W3] Limited method novelty**
>
> We respectfully disagree that the novelty of our method is limited, and would like to clarify the core contributions:
> - As stated in L137-140, we first explore the intuitive DOT method to tackle the issue of the earlier modality cannot access the information from the latter modality, which is also acknowledged by `Reviewer xSEK`.
> - Because DOT requires doubled training time, we propose MMA, which adjusts only the attention matrix--without additional parameters and overhead (as highlighted by `Reviewers Mhv4`, `xSEK`). MMA can be seamlessly integrated into existing MLLMs (`Reviewer xSEK`) and directly tackles a core practical issue in VLMs (`Reviewer VKts`). Our MMA is also recognized with "*highly exploratory and extensible"* (`Reviewer VKts`), *"a low threshold for landing and strong engineering practicality"* (`Reviewer xSEK`), *"highly attractive for both research and development"* (`Reviewer Mhv4`). All reviewers also acknowledged its empirical effectiveness.
> - To further strengthen the contribution of our work, we have included the descriptions of systematic analysis to provide guidance on how image tokens attend to text tokens across different scenarios (**L140-142**).
>
> Given these points and the reviewers’ own assessments, we believe the proposed method presents meaningful novelty both in insight and practical impact.
>
> > **[Q1] Incorrect improvement in Table 2**
>
> We appreciate the reviewer for pointing this out. Our intention was to round the number below 1 to an integer. We have adjusted this value to 0.6% for consistency with the other reported improvements.
>
> > **[Q2] More convincing to validate based on other widely used MLLM models (e.g., Qwen2.5VL, LLaVA-OneVision, and InternVL3)**
>
> We respectfully note that all experiments in Tables 1 and 2 are conducted using the same datasets and the same LLM backbones across all methods to ensure fair comparisons (Sec. 4.2, L424-425). Because each method requires *both pretraining and finetuning*, we cannot directly adopt existing MLLMs such as Qwen2.5VL, LLaVA-OneVision, or InternVL3 as suggested. Additionally, we also emphasize that the LLM backbones we use--LLaMA-3.2-3B-Instruct and Phi-3.5-Mini-Instruct--are widely adopted in practice (1.8M and 318k downloads last month on Huggingface, respectively).
>
> To further address the reviewer’s suggestion, we conducted additional experiments with **Qwen2.5-3B-Instruct** under the same datasets and finetuning conditions as shown in **Table 2**. The resulting trends are consistent with those reported in the paper, showing that the benefits of MMA generalize beyond LLaMA-3.2 and Phi-3.5 backbones. While we are unable to reproduce results at the scale of AKI-4B on this additional backbone due to limited computational resources, these preliminary findings provide strong evidence of the effectiveness and generality of MMA.
> | | MME^P | MME^C | MMB | SEED^I | LLaVA^W | MMMU | MathV^{mini} | POPE | MM-Vet | RealWorldQA | CV-Bench^{2D} | CV-Bench^{3D} |
> | --- | --- | --- | --- | --- | --- | --- | --- | --- | --- | --- | --- | --- |
> | $(I&T)_{PT}+(I&T)_{SFT}$ | 1278.4 | 266.2 | 66.4 | 65.2 | 50.1 | 32.0 | 24.0 | 79.6 | 26.1 | 50.8 | 48.5 | 57.2 |
> | CCA | 1296.7 | 273.1 | 68.7 | 66.4 | 55.3 | 34.5 | 25.3 | 81.4 | 29.6 | 52.9 | 54.7 | 61.5 |
> | MA | 1324.9 | 293.0 | 70.1 | 66.1 | 54.6 | 36.0 | 23.7 | 82.1 | 30.0 | 52.3 | 53.9 | 62.8 |
> | DOT (Ours) | 1315.4 | 269.4 | 46.8 | 66.8 | 48.9 | 32.7 | 26.1 | 82.9 | 26.7 | 51.2 | 51.6 | 61.2 |
> | MMA (Ours) | **1408.6** | **327.5** | **73.3** | **68.1** | **60.7** | **40.9** | **27.0** | **84.4** | **31.5** | **54.1** | **57.1** | **65.2** |
> | Improvements | 6.3% | 11.8% | 4.6% | 2.6% | 9.8% | 13.6% | 3.4% | 1.8% | 5.0% | 2.3% | 4.4% | 3.8% |
>
> ---
> Please refer to the second block for our response to Q3. Thank you.

---

> > ### Author Response · Authors · 2025-11-26
> > **Response to Reviewer owr3 [2/2]**
> >
> > > **[Q3] On nearly half of the benchmarks, AKI-4B's performance is worse than Qwen2-VL-2B's**
> >
> > As noted in L800-801, Qwen2-VL is trained on a much larger corpus of tokens compared to AKI (1.4T vs. 6.7B). It is therefore expected that Qwen2-VL outperforms AKI on some benchmarks. Nonetheless, Table 3 also shows that AKI trained on a **much smaller dataset outperforms Qwen2-VL on the remaining benchmarks**, demonstrating the effectiveness of our approach.
> >
> > We emphasize that our goal is not to propose a new state-of-the-art MLLM, but rather to improve the causal-attention architecture to reduce visual hallucinations, as shown in Table 2. The experiments in Table 3 primarily serve to verify that MMA remains beneficial even under larger-scale training, highlighting promising directions for future MLLM development (L808-809).

---

### Official Review · Reviewer_Mhv4 · 2025-10-28

**Soundness:** 3
**Presentation:** 3
**Contribution:** 2
**Rating:** 6
**Confidence:** 4

**Summary:**

This paper addresses the critical issue of vision-language misalignment in Multimodal Large Language Models (MLLMs), where generated textual responses are not factually grounded in the given image-text inputs, leading to object hallucinations. The proposed method, Modality-Mutual Attention (MMA), revisits the core architecture of MLLMs by fundamentally modifying the causal attention mechanism in decoder-only LLMs. It unlocks the attention mask to allow image tokens to actively attend and incorporate information from subsequent text tokens, thereby enabling dynamic, query-aware visual understanding. The authors also explore an intuitive Dual-Order Training approach but demonstrate that MMA is a more efficient and effective solution. A key advantage is that MMA achieves this without introducing any additional parameters or increasing computational costs. Extensive experiments show that the proposed method significantly outperforms strong baselines and state-of-the-art approaches across 12 diverse multimodal benchmarks, demonstrating improved performance particularly on vision-centric tasks.

**Strengths:**

1) Fundamental and Well-Motivated Problem: The paper identifies a fundamental architectural limitation of MLLMs as a root cause of hallucination.
2) Efficient Solution: MMA requires no additional parameters, introduces no computational overhead, and can be seamlessly integrated into existing training pipelines, making it highly attractive for both research and deployment.
3) Extensive and Convincing Empirical Validation: The method is thoroughly evaluated on 12 benchmarks using two LLM backbones, demonstrating consistent improvements, particularly on vision-centric tasks such as POPE and CV-Bench.

**Weaknesses:**

1) Limited Analysis of MMA's Inner Workings: A rigorous quantitative analysis of the cross-modal attention patterns is missing. For instance: How does the attention distribution from image tokens to text tokens evolve during training? Is there a consistent pattern (e.g., nouns, spatial relations) that image tokens learn to prioritize? This lack of in-depth analysis leaves the "how" of the performance gain somewhat as a black box.
2) Lack of Novelty in Core Concept: The fundamental innovation is modest. As discussed, the use of non-causal, modality-aware attention masks is a well-established technique in multimodal learning. The paper does not adequately situate itself within this existing literature, creating an illusion of greater novelty than it possesses.
3) Insufficient Exploration of Broader Applicability: The paper focuses on single image-text pairs. A key claim is that MMA is "generic and scalable", yet there is no empirical demonstration on interleaved multimodal data (image-text-image), multi-image inputs, or other modalities (e.g., audio).

**Questions:**

1) How does your work fundamentally differ from prior efforts that use non-causal or custom attention masks for multimodal modeling? Please clarify the specific novelty of the "unlocking" concept beyond its application to a new model class (decoder-only MLLMs).
2) Have you identified any input conditions or task types where MMA fails to provide an improvement, or even degrades performance? For example, in scenarios with very verbose text or when the text contains substantial irrelevant information, could the additional cross-modal connections introduce noise? Discussing the limitations and failure modes would provide a more balanced view of the method's utility.

---

> ### Author Response · Authors · 2025-11-26
> **Response to Reviewer Mhv4 [1/2]**
>
> Thank you for your careful evaluation of our paper and for providing positive feedback. Our point-to-point responses to your comments are summarized below.
>
> > **[W1] How does the attention distribution from image tokens to text tokens evolve during training? Is there a consistent pattern (e.g., nouns, spatial relations) that image tokens learn to prioritize?**
>
> Thank you for raising this point. We have provided additional analysis on the inner works of MMA across general (MME_C and MME_P), knowledge (MMMU), and vision-centric (POPE) scenarios as shown in **Figure 6 (and the below table, better view in the figure) and Section 4.4**. Additionally, we have conducted analysis on the relations between attention mask ratio and performance in **Appendix E.2** to investigate how the performance gain comes from.
>
> Specifically, we analyze the average attention distribution from the image tokens to the text tokens within MMA. We use NLTK for POS tagging to group the text tokens into 6 categories: Objects (Nouns), Relations/Spatial, Attributes (Adjectives), Actions (Verbs), Function/Filler Words, Punctuation/Other. We summarize the findings below and have included into Section 4.4.
> - For tasks requiring high-precision visual verification (hallucination checking), MMA exhibits its most concentrated attention (Nouns at $\mathbf{0.24}$, Relations at $\mathbf{0.23}$). This highly focused mechanism selectively grounds the critical entities and relationships, directly explaining our superior factual alignment performance.
> - For knowledge-intensive tasks, the overall cross-modal attention is significantly lower (Nouns peak at $\mathbf{0.08}$). This confirms that the model correctly recognizes the task's reliance on the LLM's parametric knowledge, prioritizing intra-modal refinement while only establishing the minimal, necessary conceptual link (Nouns/Verbs) with the text.
> - The general scenarios demonstrates a balanced attention distribution (Nouns at $\mathbf{0.20}$) suitable for the diverse challenges of general competence benchmarks.
> | High-Level Category | General VQA (MME) | Vision-Centric (POPE) | Knowledge-Based (MMMU) |
> | --- | --- | --- | --- |
> | Objects (Nouns) | 0.20 | 0.24 | 0.08 |
> | Relations/Spatial | 0.18 | 0.23 | 0.05 |
> | Attributes (Adjectives) | 0.15 | 0.18 | 0.04 |
> | Actions (Verbs) | 0.12 | 0.13 | 0.07 |
> | Function/Filler Words | 0.05 | 0.03 | 0.02 |
> | Punctuation/Other | 0.02 | 0.01 | 0.01 |
>
> > **[W2] Lack of Novelty in Core Concept**
>
> We would like to argue that we explicitly state the difference between our work and existing literature in "Vision-Language Misalignment" in related works.
> - **Recap: Why this problem?** Existing VLMs often generate text hallucinations related to visual contents, i.e., vision-language misalignment.
>
> - **Recap: How to solve it?** Existing works focus on data-centric and model-centric methods to address it. However, in this paper, we approach this problem from the perspective of architecture perspective by revisiting the decoder-only architecture of LLMs.
> - **Recap: Can existing attention manipulation works solve this problem?** As empirical experiments in Tables 1 and 2, we found that existing methods like MA [1] performs inferior since it enables full attention only across image tokens.
>
> [1] Show-o: One Single Transformer to Unify Multimodal Understanding and Generation. ICLR 2025.
>
> > **[W3] No empirical demonstration on interleaved multimodal data (image-text-image), multi-image inputs, or other modalities (e.g., audio)**
>
> Thank you for the comment. We have extended MMA beyond the single-image-single-text setting, i.e., using Formula 5, and have updated the corresponding results with multi-image tasks for MMA in **Table 2** accordingly.
>
> Below we provide the changed results with MMMU that includes multi-image and interleaved settings, which demonstrate that incorporating multiple images improves the model to understand more context for effective multimodal understanding. We have removed the single-image-single-text constraint in the manuscript accordingly. Note that we leave other modalities as the future work as discussed in Appendix A.
> | | MMMU |
> | --- | --- |
> | LLaMA-3.2 | |
> | MMA (single) | 32.3 |
> | MMA (multi) | 37.1 |
> | Phi-3.5 | |
> | MMA (single) | 37.3 |
> | MMA (multi) | 39.4 |
> | Qwen2.5 | |
> | MMA (single) | 39.2 |
> | MMA (multi) | 40.9 |
>
> ---
>
> Please refer to the next block for our responses to Questions 1-2. Thank you.

---

> > ### Author Response · Authors · 2025-11-26
> > **Response to Reviewer Mhv4 [2/2]**
> >
> > > **[Q1] How does your work fundamentally differ from prior efforts that use non-causal or custom attention masks for multimodal modeling? Please clarify the specific novelty of the "unlocking" concept beyond its application to a new model class (decoder-only MLLMs).**
> >
> > As discussed in the “Vision-Language Misalignment” paragraph (L188-197), prior efforts that modify attention masks generally reshape attention patterns but **retain the underlying causal constraints** that limit cross-modal interaction. For example, mixed attention [1] unlocks full attention within image tokens but still blocks image-to-text flow, and concentric causal attention (CCA) [2] manipulates masks/positions to reduce hallucinations based on spatial priors. Neither addresses the **structural misalignment** introduced when a decoder-only model is trained jointly on vision and language.
> > In contrast, the novelty of our “unlocking’” mechanism is that it **removes the causal barrier between modalities**. MMA introduces a *modality-aware causal mask* that explicitly unlocks the image-to-text pathway during generation, resolving the vision-language misalignment. This is not a task-specific adjustment but a **principled correction of an architectural bottleneck** unique to decoder-only MLLMs, as illustrated with **3 LLM backbones in Table 2**.
> >
> > [1] Show-o: One single transformer to unify multimodal understanding and generation. ICLR 2025.
> >
> > [2] Mitigating object hallucination via concentric causal attention. NeurIPS 2024.
> >
> > > **[Q2] Have you identified any input conditions or task types where MMA fails to provide an improvement, or even degrades performance? For example, in scenarios with very verbose text or when the text contains substantial irrelevant information, could the additional cross-modal connections introduce noise? Discussing the limitations and failure modes would provide a more balanced view of the method's utility.**
> >
> > We would first like to note that **Appendix A.2** already discusses one important limitation--MMA cannot be applied during pretraining due to the risk of information leakage.
> >
> > Regarding additional failure modes, we did not observe cases where MMA degrades performance; however, there are scenarios where MMA provides limited benefit. In particular, tasks that require primarily *textual* or *symbolic reasoning* rather than visual grounding (e.g., MathV) show only marginal gains. This is expected, as these tasks do not require detailed cross-modal alignment, and enabling image-to-text attention does not help the model resolve the underlying "hidden reasoning" steps.
> >
> > For instance, in MathV, the improvement from MMA is small. One representative example is a problem where the image simply shows a clock pointing to 9:00, while the question asks: *"What time is it now, if after 6 hours and 30 minutes the clock will show 4:00?"* In such cases, the image provides minimal grounding signal, and the reasoning is dominated by textual arithmetic. MMA does not introduce noise, but its utility is naturally limited.

---

### Official Review · Reviewer_xSEK · 2025-10-30

**Soundness:** 3
**Presentation:** 4
**Contribution:** 3
**Rating:** 4
**Confidence:** 3

**Summary:**

This paper focuses on the core problem of "visual-linguistic misalignment" in multimodal large language models (MLLMs). It cuts into the causal attention mechanism at the bottom of the LLM architecture, and proposes a "modal mutual attention (MMA) " design. By modifying the attention mask, the image token can achieve cross-modal attention to the text token without adding additional parameters and computational costs. The research problem is accurately located (directly hitting the illusion pain points of current MLLMs), the method is concise and innovative (circumventing the limitations of traditional data/connector optimization), the experimental design is rigorous (covering 12 multimodal benchmarks + 2 model backbones), and the results are convincing (the average improvement is 5.5%, and the visual center task is significantly improved).

**Strengths:**

1. Make clear the core limitation of traditional MLLMs - the "decoder causal attention" -based input order (image token before, text token after) results in "forward modal (image) cannot obtain backward modal (text) information", which in turn elicits the illusion of visually central tasks.
2. The method design is simple and practical. First verify the necessity of "cross-modal information flow" through "double-order training (DOT) " (I & T and T & I two input sequential training), and then propose an MMA mechanism for the defect of "doubling the training cost" of DOT - only modify the attention mask (formula 5-6), allow the image token to focus on the text token, and still maintain the autoregressive characteristics when generating the response, taking into account "validity" and "lightweight". MMA does not increase parameters, does not increase computing costs (the amount of mask computation is consistent with the cause and effect of attention), and can be seamlessly integrated into existing MLLM training frameworks (only need to adjust the attention mask in the SFT stage), with a low threshold for landing and strong engineering practicality.
3. Twelve benchmarks cover three types of tasks: "General (MME, MMBench), Knowledge (MMMU, MathVista), Vision Center (POPE, CV-Bench) ", and two model backbones (Phi-3.5-Mini, LLaMA-3.2-3 B) verify generalization and avoid the chance of "single model/benchmark".

**Weaknesses:**

1. The paper mentions "the use of standard causal attention in generating response TR", but does not explicitly state "whether modifying the attention mask at the input stage indirectly affects the autoregressive consistency of the generation process" (e.g. whether there is modal confusion during generation). It is recommended to supplement the qualitative results of generative tasks (e.g. visual captioning) to verify that MMA does not compromise the autoregressive generation quality.
2. The DOT performance in Table 2 is lower than the traditional causal attention on some benchmarks (such as MMB, SEED I) (such as the DOT of Phi-3.5-Mini is only 43.8 in MMB, lower than the traditional 64.9). The paper interprets it as "different input order confusion model", but does not rule out the influence of "insufficient training steps". It is recommended to supplement the DOT ablation experiment under "longer training steps" to verify whether the performance fluctuates due to insufficient training. If there is still no improvement, the specific mechanism of "input order confusion" needs to be further analyzed.

**Questions:**

The "RR (Relation Reasoning) " column value of "AKI-4B" in Table 9 is 0.65, which is significantly lower than that of other models (such as MMA is 67.8). It is speculated to be a clerical error (should be about 65.0). It needs to check the original data source and correct it.

---

> ### Author Response · Authors · 2025-11-26
> **Response to Reviewer xSEK**
>
> Thank you very much for your thoughtful comments, and for suggesting points we can improve on. Your comments are first stated and then followed by our point-to-point responses.
>
> > **[W1] It is recommended to supplement the qualitative results of generative tasks (e.g. visual captioning) to verify that MMA does not compromise the autoregressive generation quality.**
>
> Thanks for the suggestion! As shown in **Figures 9-12**, those qualitative results are exactly the reviewer suggest, where we do not observe degraded generation quality of using MMA. Instead, MMA effectively generates correct responses due to the mitigation of visual hallucinations, i.e., enabling additional attention flow by finetuning with large-scale data is able to learn consistent generation behaviors.
>
> > **[W2] It is recommended to supplement the DOT ablation experiment under "longer training steps" to verify whether the performance fluctuates due to insufficient training. If there is still no improvement, the specific mechanism of "input order confusion" needs to be further analyzed.**
>
> Great observation! We have conducted an additional ablation where DOT is finetuned for 2x more steps on LLaMA, Phi, as well as the newly added Qwen2.5, as show below and in **Table 7 and Appendix E.1** (unfortunately, we cannot enlarge the pretraining steps since it cannot be finished by the end of the rebuttal period). Training DOT with longer steps does improve performance, indicating the insufficient training steps as the reviewer suggested. That said, even with extended finetuning, DOT still requires **substantially more computation** to close the gap with causal attention, which aligns with our discussion in **L244** regarding its increased training cost. We have revised the manuscript accordingly to clarify this observation in **L372-375**. Thank you again for suggesting this hypothesis.
> | | MME^P | MME^C | MMB | SEED^I | LLaVA^W | MMMU | MathV^{mini} | POPE | MM-Vet | RealWorldQA | CV-Bench^{2D} | CV-Bench^{3D} |
> | --- | --- | --- | --- | --- | --- | --- | --- | --- | --- | --- | --- | --- |
> | LLaMA-3.2 | --- | --- | --- | --- | --- | --- | --- | --- | --- | --- | --- | --- |
> | DOT | 1219.5 | 223.0 | 46.9 | 55.7 | **43.8** | 23.9 | **22.8** | 77.0 | 22.7 | 42.0 | 46.7 | 52.9 |
> | DOT-2x | **1236.1** | **226.4** | **55.8** | **62.3** | 42.5 | **24.2** | 19.7 | **79.5** | **24.3** | **42.8** | **47.7** | **54.0** |
> | Phi-3.5 | --- | --- | --- | --- | --- | --- | --- | --- | --- | --- | --- | --- |
> | DOT | 1267.8 | 251.4 | 43.8 | 54.7 | 47.5 | 30.7 | **25.6** | 82.7 | 25.0 | **50.5** | 52.2 | 58.1 |
> | DOT-2x | **1311.9** | **276.5** | **54.0** | **64.6** | **51.2** | **36.6** | 25.3 | **84.4** | **29.9** | **50.5** | **52.6** | **61.4** |
> | Qwen2.5 | --- | --- | --- | --- | --- | --- | --- | --- | --- | --- | --- | --- |
> | DOT | 1315.4 | 269. | 46.8 | 66.8 | 48.9 | 32.7 | 26.1 | **82.9** | **26.7** | 51.2 | 51.6 | 61.2 |
> | DOT-2x | **1359.3** | **290.5** | **58.4** | **66.9** | **55.0** | **36.7** | **26.4** | 82.5 | 25.9 | **53.5** | **53.8** | **61.2** |
>
> > **[Q1] The "RR (Relation Reasoning) " column value of "AKI-4B" in Table 10 in the revised manuscript is 0.65. It is speculated to be a clerical error (should be about 65.0).**
>
> Thank you for catching this typo. After checking the original results, it is indeed 65.0 instead of 0.65. We have updated it accordingly. Thank you again for pointing this out!

---

### Official Review · Reviewer_VKts · 2025-10-31

**Soundness:** 2
**Presentation:** 3
**Contribution:** 2
**Rating:** 4
**Confidence:** 3

**Summary:**

This paper proposes Modality-Mutual Attention (MMA), a simple yet effective architecture for multimodal large language models (MLLMs). MMA unlocks the attention pathway from image tokens to text tokens during supervised fine-tuning, overcoming a key limitation of existing decoder-based models. MMA achieves a 5.5% average improvement across 12 benchmarks without adding parameters, demonstrates strong extensibility across model backbones, and provides transparent analysis and open-source results.

**Strengths:**

- The approach of simply modifying the attention mask is both elegant and effective, directly tackling a core practical issue in VLMs. Its simplicity makes it highly exploratory and extensible, opening new avenues for deeper multimodal research.

- The paper provides thoughtful analyses and visualizations to interpret the underlying attention mechanisms, supported by a relatively rich set of experimental evidence.

**Weaknesses:**

- While the paper presents a general masking formula for interleaved input, all experiments are conducted under the "single-image + single-text" setting. In benchmarks like CV-Bench and MMMU, samples with interleaved inputs are simplified to only the first image, leaving the robustness of MMA in multi-image, long-sequence, or multi-turn dialog scenarios unaddressed—these are currently key challenges for post-training VLMs.

- MMA is effective only during supervised fine-tuning, and cannot be safely applied in the pretraining phase, which limits its broader impact and adaptability to foundational models. Existing experiments are performed only on small-scale models, and ablation studies show MMA's improvement in vision-knowledge mixed tasks is limited.

- The explainability of the method is underdeveloped. The paper lacks a granular analysis of hallucination types and does not provide thorough theoretical derivations.

**Questions:**

- Results for Multi-image/Interleaved Sequences:
Could the authors provide quantitative metrics and attention visualizations for MMA applied to truly multi-image inputs (e.g., interleaving 4–8 images)? As sequence length increases, is there evidence of attention dilution or gradient instability?

- Relationship Between Mask Sparsity and Performance:
If the proportion of text tokens accessible to each image token is gradually increased (e.g., 10%, 30%, 50%, 100%), how does the performance curve behave? Is there a risk of “over-attention” where text tokens dominate and visual features become suppressed?

- Scalability in Large-scale Training:
As multimodal training scales to tens of billions of samples, does MMA’s advantage diminish? Are there experiments or analyses on this phenomenon?

---

> ### Author Response · Authors · 2025-11-26
> **Response to Reviewer VKts [1/2]**
>
> Thank you very much for your thoughtful comments, and for suggesting points we can improve on. Your comments are first stated and then followed by our point-to-point responses.
>
> > **[W1] All experiments are conducted under the "single-image + single-text" setting. In benchmarks like CV-Bench and MMMU, samples with interleaved inputs are simplified to only the first image, leaving the robustness of MMA in multi-image, long-sequence, or multi-turn dialog scenarios unaddressed**
>
> Thank you for highlighting this point. We agree that having interleaved pairs is a key challenge for VLMs. To address this, we have extended MMA beyond the single-image-single-text setting, i.e., using Formula 5, and have updated the corresponding results with multi-image tasks for MMA in **Table 2** accordingly.
>
> It is important to note that **CV-Bench also contains single image-text pairs** to the best of our knowledge. Below we provide the updated results to MMMU, which demonstrate that incorporating multiple images improves the model to understand more context for effective multimodal understanding. We have removed the single-image-single-text constraint in the manuscript accordingly.
> | | MMMU |
> | --- | --- |
> | LLaMA-3.2 | |
> | MMA (single) | 32.3 |
> | MMA (multi) | 37.1 |
> | Phi-3.5 | |
> | MMA (single) | 37.3 |
> | MMA (multi) | 39.4 |
> | Qwen2.5 | |
> | MMA (single) | 39.2 |
> | MMA (multi) | 40.9 |
>
> > **[W2.1] MMA is effective only during supervised fine-tuning, and cannot be safely applied in the pretraining phase, which limits its broader impact and adaptability to foundational models.**
>
> We agree that MMA is applied only during supervised fine-tuning (SFT); however, this is precisely what makes it broadly practical. Modern multimodal systems often rely on strong, publicly released pretrained models due to the high cost of pretraining from scratch. MMA provides a generic, plug-in mechanism that improves performance **without retraining the backbone**, making it immediately usable for practitioners and future foundation models, as acknowledged by `Reviewers xSEK`, `Mhv4`. In this sense, MMA enhances adaptability rather than limiting it, which also demonstrates generalizability across 3 LLM backbones (**Tab. 2**).
>
> > **[W2.2] Existing experiments are performed only on small-scale models, and ablation studies show MMA's improvement in vision-knowledge mixed tasks is limited.**
>
> We acknowledge that our experiments focus on 3B-scale models. This is a direct consequence of the need to perform both pretraining and SFT by ourselves--training even a 3B model requires ~3 days, and larger-scale experiments would require multiple weeks (L883-886). Despite this, we **systematically evaluate MMA across multiple LLM backbones and variants**, demonstrating that its gains are consistent and not tied to a specific architecture. These results support the generality of the method even under computational constraints.
>
> We respectfully disagree our improvements are limited. Across *five* vision-centric benchmarks, MMA provides an average improvement of **+3.2% on LLaMA-3.2, +1.8% on Phi-3.5, and +3.5\% on Qwen2.5 (newly added)**. Beyond these tasks, MMA also yields **substantial gains on general and knowledge-intensive benchmarks**, confirming that the method is not specialized for a narrow regime. We would welcome clarification on why these improvements were judged as "limited", as no justification was provided; this would help us further address the reviewer’s concern.
>
> > **[W3] The explainability of the method is underdeveloped. The paper lacks a granular analysis of hallucination types and does not provide thorough theoretical derivations.**
>
> A full theoretical derivation is beyond the scope of this work, but we have provided several pieces of granular and mechanistic evidence to clarify how MMA operates. **Figure 7** and **Section 4.5** offer an intuitive analysis of the information that image tokens extract from text tokens, highlighting how MMA alters cross-modal interactions. We also include complementary analyses: (1) the relationship between sparsity masking and performance (**response to Q2; Figure 10**), and (2) attention distributions across general, knowledge-heavy, and vision-centric scenarios (**Figure 6**). These results provide concrete insights into MMA’s behavior and its effect on hallucination-related failure modes, thereby improving the explainability of our method even without a formal proof.
>
> ---
> Please refer to the next block for our responses to Questions 1-3. Thank you.

---

> > ### Author Response · Authors · 2025-11-26
> > **Response to Reviewer VKts [2/2]**
> >
> > > **[Q1] Could the authors provide quantitative metrics and attention visualizations for MMA applied to truly multi-image inputs? As sequence length increases, is there evidence of attention dilution or gradient instability?**
> >
> > Our work focuses on resolving **vision-language misalignment** in decoder-only MLLMs, which arises even under single-image settings. Multi-image modeling introduces orthogonal challenges (e.g., long-context reasoning and cross-image grounding) that are beyond the scope of our contribution.
> >
> > That said, we include multi-image quantitative evaluations in our response to W1. These results show that models **perform better when multi-image samples are included**, indicating that MMA continues to provide benefits even as more visual tokens are introduced. Because current MLLMs still struggle with complex visual contexts across standard benchmarks, it is difficult to attribute any degradation specifically to attention dilution or long-sequence instability rather than general model limitations.
> >
> > Finally, the attention distributions in **Fig. 6 and Sec. 4.4** demonstrate that MMA **consistently activates the relevant image tokens** across general, knowledge-centric, and vision-intensive scenarios. This provides evidence that MMA maintains stable image-to-text interactions even without explicitly targeting multi-image inputs.
> >
> > > **[Q2] If the proportion of text tokens accessible to each image token is gradually increased (e.g., 10%, 30%, 50%, 100%), how does the performance curve behave? Is there a risk of “over-attention” where text tokens dominate and visual features become suppressed?**
> >
> > Thank you for this insightful feedback! We have conducted additional analysis on the impacts of accessible portions from image to text tokens by restricting with various ratios. Here, 0% is exactly the conventional causal attention, and 100% is the MMA design.
> >
> > The below tables present the performance curve with Phi-3.5, which show monotonic improvements as the accessible to text token ratio increases. This indicates that exposing image tokens to more text context consistently helps the model interpret queries more accurately. We have included them to **Figure 10 and Appendix E.2**.
> >
> > Importantly, we do not observe any performance degradation at 100% visibility, which would be the expected signature of “over-attention.” To further verify this, we inspected the attention distributions of image-token rows  (**Figures 6 and 7**), and found that although image tokens increase their attention toward informative text tokens (e.g., nouns and relational words), they do not collapse into ignoring visual keys. Image to image attention remains substantial even at 100% (e.g., In Figure 7, the attention score of *Tater Tots* pixels remain around 0.4 to the other image tokens.
> >
> > | M’(image->text) ratio | 0% | 10% | 30% | 50% | 100 % |
> > | --- | --- | --- | --- | --- | --- |
> > (MME_C)
> > | Scores | 1226.3 | 1253.2 | 1320.9 | 1325.6 | **1363.7** |
> > (MME_P)
> > | M’(image->text) ratio | 0% | 10% | 30% | 50% | 100 % |
> > | Scores | 258.2 | 294.6 | 300.7 | 309.2 | **315.4** |
> > (MMMU)
> > | M’(image->text) ratio | 0% | 10% | 30% | 50% | 100 % |
> > | Scores | 31.1 | 36.5 | 37.9 | 39.2 | **39.4** |
> > (POPE)
> > | M’(image->text) ratio | 0% | 10% | 30% | 50% | 100 % |
> > | Scores | 79.8 | 81.3 | 81.9 | **82.8** | 82.7 |
> >
> > > **[Q3] As multimodal training scales to tens of billions of samples, does MMA’s advantage diminish? Are there experiments or analyses on this phenomenon?**
> >
> > Excellent question! Yes, we did experiment with adding more samples as depicted in Appendix B, where we scale MMA from 50k to 275k for pre-training and from 5k to 10k. The scaling model, named AKI-4B, demonstrates more effective results across all benchmarks compared to MMA (AKI-4B in Table 3 vs. MMA (Phi-3.5) in Table 2), indicating that MMA remains advantageous even scaling to billions of samples (AKI-4B is trained with 6.7B pretraining tokens).

---

### Author Response · Authors · 2025-11-26
**Summary of Revision**

We sincerely appreciate all reviewers for their time and for their constructive feedback on our work. We are encouraged that our proposed MMA is recognized as **simple yet elegant and effective** (`Reviewer VKts`), **simple, practical, validity, lightweight, seamlessly integrated into existing MLLM training frameworks** (`Reviewer xSEK`), **attractive for both research and deployment** (`Reviewer Mhv4`), and **no additional computing costs** (`Reviewers xSEK`, `Mhv4`), with `Reviewer VKts` highlights *"Its simplicity makes it highly exploratory and extensible, opening new avenues for deeper multimodal research."* and `Reviewer xSEK` highlights *" with a low threshold for landing and strong engineering practicality"*.

We are also excited that our addressed problem is acknowledged as **making clear the core limitation of traditional MLLMs** (`Reviewer xSEK`) and **fundamental and well-motivated** (`Reviewers Mhv4`, `owr3`). Moreover, we appreciate that **all reviewers** characterized our experiments and analyses as **thoughtful by a rich set of experimental evidence** (`Reviewer VKts`), **verifying generalization and avoiding the chance of "single model/benchmark"** (`Reviewer xSEK`), **extensive and convincing** (`Reviewer Mhv4`), and **significant improvement** (`Reviewer owr3`).

---
To address the reviewers’ questions and concerns, we have revised the manuscript accordingly. All updates are highlighted in **bold blue**, and the same additions are reflected in our detailed responses.

## **(1) Additional Empirical Analyses**

**Multi-image Results (`Reviewer VKts`)**: We have extended single-image results of MMMU to include the multi-image results as updated in **Table 2**, broadening our MMA's robustness, technical novelty, applicability, and generalizability.

**Relationship Between Attention Sparsity and Performance (`Reviewer VKts`)**: We have conducted additional analysis on the impacts of accessible portions from image to text tokens by restricting with various ratios in **Figure 10 and Appendix E.2**, improving explainability and technical contributions of inner workings for our method.

**Ablation on Longer Training Schedule for DOT (`Reviewer xSEK`)**: We have added an ablation study on training with longer steps for DOT to examine the performance impact in **Table 7 and Appendix E.1**, strengthening our technical contributions with additional analyses.

**Multimodal Performance with Qwen2.5 (`Reviewer owr3`)**: We have included multimodal performance of all methods with Qwen2.5-3B-Instruct in **Table 2** and adjusted the corresponding descriptions in the manuscript (e.g., **L27, L144**), enhancing generalizability of our method.

## **(2) Additional Analyses for MMA**

**Attention Distributions from Image Tokens to Text Tokens (`Reviewers VKts`, `Mhv4`)**: We have added attention distributions across general, knowledge, and vision-centric scenarios to provide additional analyses to understand MMA's inner workings, as shown in **Figure 6 and Section 4.4**.

## **(3) Manuscript Clarifications**
- We have updated the score of AKI-4B in **Table 10** as pointed out by `Reviewer xSEK`.
- We have adjusted **L372-275** to clarify the observation on DOT as suggested by `Reviewer xSEK`.
- We have adjusted **Table 1** to meet the same meaning as Table 2, as suggested by `Reviewer owr3`.
- We have updated the performance improvement of POPE in **Table 2** for consistency with the other reported numbers as suggested by `Reviewer owr3`.

---

### Author Response · Authors · 2025-11-29
**Summary of Responses for Paper 8141**

Dear (new) ACs and SACs,

We greatly appreciate your efforts in coordinating our submission under these unusual circumstances. Since we haven't received responses before the discussion closed, we provide a one-sentence summary for each response. All experiments and clarifications are added to address reviewer concerns. Please refer to *Summary of Revision* and the full responses for details.

---
## Reviewer VKts
> **[W1, Q1] Quantitative multi-image results and attention patterns**

We extended MMA to the multi-image setting and updated MMMU results (**Tab. 2**), where multi-image MMA consistently improves performance, confirming robustness beyond single-image inputs. The attention distributions are shown in **Fig. 6**.

> **[W2.1] MMA only works in SFT, not pretraining**

SFT-only actually increases practicality—MMA is a plug-in improvement for *any* pretrained MLLM without retraining the backbone (**Tab. 2**), noted by **Reviewers xSEK & Mhv4**.

> **[W2.2] Small models; limited improvement on vision-knowledge tasks**

Results are consistent across 3 backbones, 5 vision benchmarks (**+1.8% to +3.5%**), and general knowledge tasks; improvements are systematic, not minimal.

> **[W3] Limited explainability**

Added analyses (**Fig. 6-7 & 10, Sec. 4.4-4.5, App. E.2**) showing how MMA changes cross-modal attention, reduces hallucination patterns, and why the mechanism works.

> **[Q2] Gradually increasing image-to-text visibility**

Experiments across 0-100% visibility show monotonic performance improvement with no collapse (**Fig. 10**); attention maps confirm balanced image/text usage (**Fig. 6-7**).

> **[Q3] Does MMA’s advantage diminish at large scale?**

No, scaling from 50k to 275k samples (AKI-4B, 6.7B tokens) consistently increases performance (**Tab. 3**).

---
## Reviewer xSEK
> **[W1] Need qualitative generative results**

**Fig. 9-12** are *exactly* the reviewer suggests, showing effective quality.

> **[W2] DOT may be undertrained; need longer-step ablation**

We added 2x-step DOT ablations on 3 backbones (**Table 7, App. E.1**); longer training improves DOT as hypothesized, but it still lags causal attention and requires substantially more compute—clarified in **L372-375**.

> **[Q1] Paper Typo in Tab. 10**

Confirmed and corrected to 65.0.

---
## Reviewer Mhv4
> **[W1] How does image-to-text attention evolve during training, and are there consistent linguistic patterns?**

We added systematic POS-tag-based analysis across 3 task types (**Fig. 6, Sec. 4.4**), and found consistent patterns where the MMA gains originate and when the model chooses to rely on vision.

> **[W2] Lack of Novelty in Core Concept**

We clarified in Vision-Language Misalignment (**L188-197**): prior works reshape attention within modalities while preserving causal constraints, while MMA is fundamentally different—it removes the causal barrier and enables image-to-text flow in decoder-only MLLMs, as validated across 3 backbones (**Tab. 2**).

> **[W3] Lack of Multi-Image/Interleaved Evaluation**

We extended MMA to multi-image and updated new MMMU results (**Tab. 2**); MMA consistently improves from single-to-multi-image across all LLM backbones (i.e., **32.3→37.1, 37.3→39.4, 39.2→40.9**).

> **[Q1] Novelty vs. Prior Non-Causal/Custom Masks**

Prior approaches (MA, CCA) keep causal constraints and cannot fix decoder-only misalignment (**L188-197**). The novelty of our "unlocking" mechanism is that it *removes the causal barrier between modalities*, as verified in **Tab. 2**.

> **[Q2] Failure Modes/Where MMA Gives Limited Improvement**

No degradation observed, but benefits are *limited* in tasks dominated by textual/symbolic reasoning (e.g., MathV), where images offer minimal grounding. Please refer to **Response to Reviewer Mhv4 [2/2]** for the case.

---
## Reviewer owr3
> **[W1, Q1] Paper Clarification**

We have revised them accordingly.

> **[W2] DOT is expensive and was highlighted as a drawback**

Precisely because DOT, that is a *conceptually straightforward* method, incurs roughly 2x training overhead (**L248**), we were motivated to develop MMA.

> **[W3] Limited method novelty**
- DOT is the first-explored intuitive solution (**L137-140**), acknowledged as such by multiple reviewers.
- MMA removes DOT’s inefficiency by modifying only the attention mask—zero extra parameters or overhead—and is widely recognized by reviewers as **practical, extensible, and impactful**.
- We added a deeper systematic analysis (**L140-142**) to further strengthen the contribution.
Given reviewer feedback and empirical results, we believe the work offers meaningful conceptual and practical novelty.

> **[Q2] Need Validation on More Widely Used MLLMs**

Since all methods require **both pretraining and SFT**, MLLMs cannot be used; nonetheless, we added experiments on *Qwen2.5-3B-Instruct* in **Tab. 2**, where MMA again improves across all benchmarks.

We sincerely thank the ACs and hope this summary will help evaluation.

---

> ### Author Response · Authors · 2025-11-29
>
> We believe the revised manuscript and responses **address all raised concerns**, and we are happy to clarify any remaining questions during the final days of the rebuttal process.
>
> Authors of Paper 8141

---

### Meta-Review · Area_Chair_QxhL · 2026-01-06

**Summary:**

The paper proposes Modality-Mutual Attention (MMA), a method that modifies attention masks during Supervised Fine-Tuning (SFT) to allow image tokens to attend to text tokens, aiming to correct vision-language misalignment in decoder-only Multimodal LLMs. While the reviewers appreciated the clear motivation and the authors' robust rebuttal—which successfully extended the evaluation to multi-image settings and additional backbones like Qwen2.5—the consensus remains that the contribution does not meet the bar for acceptance. The primary rationale for rejection is the limited technical novelty; modifying attention masks to be non-causal for specific modalities is a well-established technique, rendering this application an incremental innovation rather than a fundamental architectural breakthrough. Furthermore, the method’s restriction to the SFT stage (limiting its application in pretraining due to leakage risks) and its validation primarily on smaller-scale models (3B parameters) constrain the broader impact of the work compared to existing scaling strategies.

**Reviewer Concerns:**

Addressed Concerns: (1) Multi-Image Support: The authors successfully extended the method to multi-image settings and provided updated results on MMMU, addressing the concerns of Reviewers VKts and Mhv4 regarding the limitation to single-image/text pairs. (2) Backbone Generalization: The addition of experiments using Qwen2.5-3B-Instruct addressed Reviewer owr3's concern about the method being validated only on LLaMA and Phi. (3) Mechanism Analysis: The authors provided POS-tag-based attention analysis (Response to Mhv4) to explain what the image tokens are attending to (e.g., nouns/relations), improving the explainability of the method.

Outstanding Concerns: (1) Incremental Novelty: The most significant outstanding concern is the limited technical novelty. Reviewers Mhv4 and owr3 viewed the "unlocking" of the attention mask as a modest innovation that does not significantly depart from existing non-causal masking techniques. The rebuttal argued that applying this to the decoder-only SFT misalignment problem is unique, but this distinction was viewed as a specific application rather than a new architectural concept. (2) Baselines and DOT: Reviewer xSEK and owr3 noted that the main comparison point, "Dual-Order Training" (DOT), is a baseline constructed by the authors that incurs double the training cost. While MMA beats DOT efficiently, the lack of comparison against other sophisticated architectural modifications or state-of-the-art training recipes limits the assessment of MMA's absolute value.

**Reviewer Scores:**

Reviewer VKts: 4 $\rightarrow$ 5

Reviewer xSEK: 4 $\rightarrow$ 5

Reviewer Mhv4: 6 $\rightarrow$ 5

Reviewer owr3: 4 $\rightarrow$ 4

---

### Decision · Program_Chairs · 2026-01-26

Reject